



# Improving solution availability and temporal consistency of an optimal estimation physical retrieval for ground-based thermodynamic boundary layer profiling

Bianca Adler [1,2], David D. Turner[3], Laura Bianco[1,2], Irina V. Djalalova[1,2], Timothy Myers[1,2], and James M. Wilczak[2]

[1]CIRES, University of Colorado Boulder
[2]NOAA Physical Sciences Laboratory
[3]NOAA Global Systems Laboratory

**Correspondence:** Bianca Adler  (bianca.adler@noaa.gov)

**Abstract.** Thermodynamic profiles in the atmospheric boundary layer can be retrieved from ground-based passive remote sensing instruments like infrared spectrometers and microwave radiometers with optimal estimation physical retrievals. With a high temporal resolution on the order of minutes, these thermodynamic profiles are a powerful tool to study the evolution of the boundary layer and to evaluate numerical models. In this study, we present three recent modifications to the Tropospheric

Remotely Observed Profiling via Optimal Estimation (TROPoe) retrieval framework, which improve the availability of valid solutions for different atmospheric conditions and increase the temporal consistency of the retrieved profiles. The characterization of the uncertainty of the input and the choice of spectral infrared bands is crucial for retrieval performance and we present methods to enhance the availability of valid solutions retrieved from infrared spectrometers by preventing overfitting and by adding information from an additional spectral band in high moisture environments. Since each profile is retrieved indepen-

dently from the previous one, the time series of the thermodynamic variables contain random uncorrelated noise, which may hinder the study of diurnal cycles and temporal tendencies. By including a previous retrieved profile as input to the retrieval, we increase the temporal consistency between subsequent profiles without suppressing real mesoscale atmospheric variability. We demonstrate that these modifications work well at mid-latitudes, polar, and tropical sites and for retrievals based on infrared spectrometers and microwave radiometers measurements.

## 1  Introduction

For the analysis of physical processes in the atmospheric boundary layer (ABL), the evaluation of numerical weather prediction models, and data assimilation, observations of the continuous temporal evolution and diurnal cycle of thermodynamic profiles are essential (National Research Council, 2009; Wulfmeyer et al., 2015). Airborne platforms such as radiosondes, tethersondes, crewed aircraft, or uncrewed aircraft systems usually do not measure continuously and only provide snapshots of the atmo-

spheric conditions. Ground-based active remote sensing instruments such as Raman lidar (e.g., Turner et al., 2016; Di Girolamo et al., 2017) and differential absorption lidar (DIAL) (e.g., Newsom et al., 2020; Spuler et al., 2021) or passive remote sens-



ing instruments such as infrared spectrometers (IRS) (Turner and Löhnert, 2014) or microwave radiometers (MWR) (Crewell and Löhnert, 2007; Löhnert et al., 2009) can resolve rapid temporal changes of the thermodynamic ABL state. The benefit of networks of ground-based profiling instruments for operational and research purposes has been outlined in several previous

works (Löhnert and Maier, 2012; De Angelis et al., 2017; Illingworth et al., 2019; Wagner et al., 2019; Cimini et al., 2020; Degelia et al., 2020; Shrestha et al., 2021; Böck et al., 2024). While active sensors emit and receive electromagnetic waves, passive sensors detect the emitted radiance of the atmosphere in certain spectral regions from which atmospheric variables have to be retrieved. One option to retrieve thermodynamic profiles from these passively sensed radiances is based on the optimal estimation approach (Maahn et al., 2020), which combines measurements, prior information, and corresponding uncertainties.

Based on the AERIoe optimal-estimation physical retrieval algorithm (Turner and Löhnert, 2014), which only allowed infrared radiances as input, the Tropospheric Remotely Observed Profiling via Optimal Estimation (TROPoe, Turner and Blumberg, 2019; Turner and Löhnert, 2021) retrieval was developed. TROPoe allows combining radiances observed by MWR or IRS along with thermodynamic profiles from various sources such as Raman lidar (Turner and Blumberg, 2019), DIAL (Turner and Löhnert, 2021), Radio Acoustic Sounding Systems (RASS) (Djalalova et al., 2022), radiosondes, or numerical weather

prediction models (Bianco et al., 2024). After more than 10 years of development, the TROPoe retrieval code was recently converted to Python and put into a Docker container to facilitate its usage for both operations and research. It is currently used operationally by the Department of Energy (DOE) Atmospheric Radiation Measurement (ARM) program (Turner and Ellingson, 2016) and by the Swiss weather service MeteoSwiss as part of the EUMETNET (Rüfenacht et al., 2021). TROPoe can be used to process data from a network of inhomogeneous thermodynamic profiling instruments consisting of different

passive and active remote sensors. Its uniform data output, which includes a full error characterization, information content, and vertical resolutions, facilitates subsequent analysis and data assimilation of ground-based profiling observing networks.

TROPoe determines the optimal state vector which may consist of thermodynamic profiles as well as cloud and trace gas properties which satisfies both the observations and the climatological information (the prior). The prior is needed to constrain the ill-posed retrieval to realistic solutions and specifies how temperature and humidity covary with height. Starting with

the prior as a first guess, a forward model is used to compute pseudo-observations, which are then compared to the actual observations. If the computed and observed values do not agree within the uncertainty of the measurements, this process is iteratively repeated.

The development of TROPoe started in 2011 and is still ongoing. Various modifications, improvements, and evaluations have been described in previous papers (Blumberg et al., 2015; Turner and Blumberg, 2019; Turner and Löhnert, 2021; Djalalova

et al., 2022). In the present paper, we address three specific issues related to (i) adequately characterizing uncertainties of the input, (ii) improving availability of valid solutions in high moisture environments, and (iii) improving the temporal consistency of retrieved thermodynamic profiles. The first two issues are IRS specific, while the third issue applies to both MWR and IRS-based retrievals.

1. Ideally, uncertainties in the observations, prior, and forward model are propagated and characterized by the posterior
covariance matrix which is part of the TROPoe output. Because including the uncertainty of the forward model would increase the computational costs of the retrieval substantially, the uncertainty of the forward model is assumed to be



zero in the current framework of TROPoe. To compensate for the missing uncertainty of the forward model and to prevent overfitting of the data for IRS-based retrievals, the random noise associated with the observed infrared radiances is inflated. This uncertainty is instrument specific and is determined during the IRS calibration process (see Revercomb
et al. (1988) and Knuteson et al. (2004b) for details). Before IRS data are used within TROPoe, the random noise is often greatly reduced using a principal component-based noise filter (Turner et al., 2006). By using the radiance uncertainty before noise filtering together with the noise filtered radiances in the retrieval, the higher radiance uncertainty is intended to compensate for the missing forward model uncertainty. For details on this approach see Turner and Blumberg (2019). However, depending on the radiance noise level of a specific IRS, this might not be sufficient and may still lead to
overfitting of the data and unrealistic profiles (Adler et al., 2023). We propose a minimum noise level which should be used for the IRS radiances in TROPoe as an intermediate solution before a computationally efficient implementation of the IRS forward model error can be included in the TROPoe framework. Because the signal to noise ratio in the MWR brightness temperature observations is lower than for the IRS radiances, overfitting is less of an issue for MWR-based retrievals.

2. TROPoe traditionally uses spectral information between 538-588 cm$^{-1}$ to retrieve water vapor and 612-722 cm$^{-1}$ to retrieve temperature from IRS measurements. The information content in the water vapor profile retrieval decreases as the precipitable water increases (Turner and Löhnert, 2014, their Fig. 7). This can be explained by the saturation of these spectral bands in moist environments as illustrated in Fig. 1 for a relatively high (red curve) and low (gray curve) moisture environment. The spectral bands most sensitive to water vapor (green shading) are saturated and contain little
information content in the moist environment. In addition to having less information content, the retrieval often struggles to converge and to provide a valid solution. We investigate how adding an additional spectral band from 793-804 cm$^{-1}$ (hatched green band labeled WVBAND in Fig. 1) in situations where the traditional water vapor bands are saturated can increase the information content and help the retrieval to find a valid solution.

3. In the current TROPoe framework, every timestamp is processed separately without using any information from the
previous state of the atmosphere. This leads to noisy time series (within the uncertainty limits of the retrieval), which frequently manifests as vertical stripes in time-height cross sections (e.g., Turner and Löhnert, 2014, their Fig. 9; Turner and Blumberg, 2019, their Fig. 13). An additional limitation for IRS-based retrievals is that the spectrum starts to become opaque for clouds with liquid water vapor values above around 60 g m$^{-2}$ (Turner, 2007). In the presence of such clouds, the information above cloud base hence comes mostly from the climatological prior or other sources such as model
and radiosonde data. Profiles in the presence of short-lived cumulus clouds thus suffer from low temporal consistency limiting their benefit for the analysis of diurnal cycles (e.g., Turner and Blumberg, 2019, their Fig. 13). We investigate how including information from previous retrieved thermodynamic profiles as input to the retrieval can increase temporal consistency of the profiles, potentially enhancing their value for the analysis of physical processes and diurnal cycles.

To address these three issues in TROPoe, we ran different experiments for IRS and MWR measurements and evaluated the
results against collocated radiosonde thermodynamic profiles and Raman lidar water vapor mixing ratio profiles. In an attempt





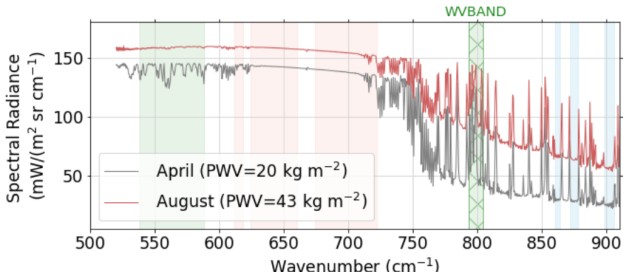

**Figure 1.** Downwelling spectral radiances as observed by IRS at SGP on a relatively dry day (April 21 2019) and on a moist day (August 7 2019). Precipitable water vapor (PWV) for each day is given in parentheses. Shading indicates spectral bands used in TROPoe with green being primarily sensitive to water vapor, red to temperature, and blue to clouds. The hatched band indicated with WVBAND is the additional band proposed in this work for better retrieval performance in moist environments.

to generalize the findings, we tested the experiments for measurements in different climatological regimes and utilized data from mid-latitude, tropical, and polar sites.

The manuscript is structured as follows: Section 2 describes the used sites and instrumentations as well as the TROPoe retrieval framework. In Sect. 3, the TROPoe experiments are introduced and in Sect. 4 the impact of the different experiments 95 on the thermodynamic profiles are analyzed.

## 2 Sites, instrumentation, and retrieval

### 2.1 Sites and instrumentation

The evaluation of the different TROPoe experiments requires sites with MWR and/or IRS measurements as well as regular radiosonde launches. Table 1 provides information on the sites and available sensors. We utilize data from the Summit Station 100 site (SMT, Shupe et al., 2013) supported by the National Science Foundation (NSF) in Greenland as an example for a polar site, the ARM Manacapuru site (MAO, Martin et al., 2017) in Brazil for a tropical site, and the ARM Southern Great Plains site (SGP, Sisterson et al., 2016) in Oklahoma in the United States for a mid-latitude site. At these three sites, a multi-channel MWR for thermodynamic profiling is deployed only at SMT. To evaluate the MWR-based retrievals for a mid-latitude site, we hence use data from the Lindenberg site (LIN) in Germany where a MWR was operated for the FESSTVAL field campaign 105 in 2021 (Hohenegger et al., 2023). As an example for a tropical site, we use MWR data from the Save site (SAV) in Benin in south-western Africa where a MWR was deployed for the DACCIWA field campaign in 2016 (Kalthoff et al., 2018; Kohler et al., 2022).

The thermodynamic conditions at the different sites are substantially different and spanned a wide range of temperature and water vapor mixing ratio values illustrated by the near-surface measurements (Fig. 2). The polar site SMT is characterized by 110 below freezing temperature and water vapor mixing ratios of less than 2 g kg$^{-1}$. The conditions at the mid-latitudes sites, LIN





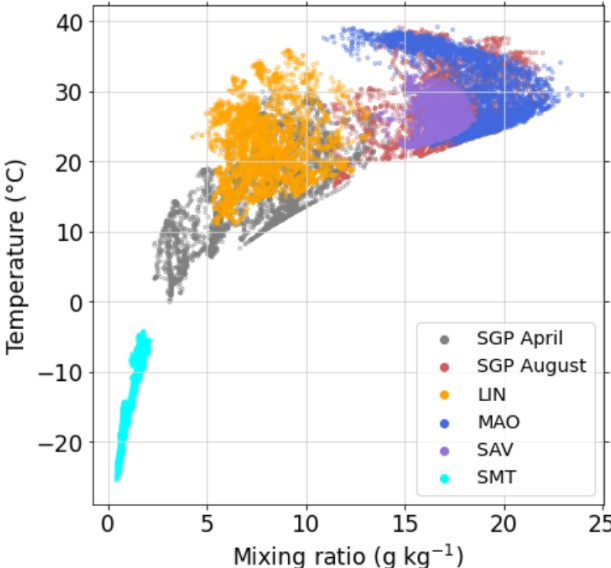

**Figure 2.** Near-surface temperature versus near-surface water vapor mixing ratio during the investigated periods at the different sites.

and SGP in April, are very similar with temperature values between 0 and 30 °C and water vapor mixing ratio mostly below 12 g kg$^{-1}$, while SGP in August resembles the two tropical sites MAO and SAV with temperatures above 20 °C and water vapor mixing ratio often higher than 15 g kg$^{-1}$.

The IRS instruments used in this study are Atmospheric Emitted Radiance Interferometers (AERI, Knuteson et al., 2004a, b).
The AERI retrieves downwelling infrared radiation between the wavelengths of 3.3 and 19 $\mu m$ (between the wavenumbers of 520 to 3,000 cm$^{-1}$) at a spectral resolution of about 0.5 wavenumber. The spectral bands used in TROPoe are indicated in Fig. 1. The IRS radiances were noise filtered using the principal component analysis (Turner et al., 2006) and a spectral calibration was applied following the method described in (Knuteson et al., 2004b). The IRS has a hatch that closes during precipitation events to protect the fore optics, which inhibits measurements during rain or snow.

The MWR instruments used are Humidity and Temperature Profilers (HATPRO, Rose et al., 2005) and measure microwave radiation in 14 channels with 7 channels being distributed around the 22.2 GHz water vapor absorption line and the other 7 along the low frequency wing of the oxygen absorption complex at 60 GHz. Several times per hour the MWRs performed low-elevation angle scans to increase the information content of temperature profiles in the boundary layer (Crewell and Löhnert, 2007).

More than 100 radiosonde profiles were available during the month-long periods at SGP, MAO, and LIN where radiosondes were launched 4 times per day (Tab. 1). About 60 radiosondes were available at SMT and SAV. Radiosondes were launched twice per day at SMT and once daily and during intensive observation periods at SAV. Note that the numbers of radiosonde profiles used for the evaluation of the TROPoe retrievals is lower than the maximum available number, because we only considered profiles under clear-sky conditions when all TROPoe experiments provided valid solutions for the evaluation.




**Table 1.** Overview of sites, instruments, and periods used to test the different TROPoe experiments. Station height is given in m above mean sea level (m MSL). Note that the number of radiosonde profiles used for the evaluation of TROPoe retrievals (in brackets) is lower than the total number of launched radiosondes.

| Site ID | Site name | Geographic location and height | Instrument | Period | Number of radiosondes (total (clear sky and time-matched) | Characteristic | Comment |
|---|---|---|---|---|---|---|---|
| SGP | Southern Great Plains | 36.61° N, -97.48° E, 237 m MSL | IRS | April 2019 | 146 (75) | mid-latitude | Raman lidar available |
| | | | | August 2019 | 109 (26) | | |
| MAO | Manacapuru | -3.21° N, -60.60° E, 50 m MSL | IRS | September 2015 | 118 (68) | tropical | |
| SMT | Summit Station | 72.60° N, -38.43° E, 3255 m MSL | IRS  MWR | June 2015 | 61 (39) | polar | |
| LIN | Lindenberg | 52.21° N, 14.12° E, 98 m MSL | MWR | June 2021 | 122 (102) | mid-latitude | |
| SAV | Save | 8.00° N, 2.10° E, 166 m MSL | MWR | July 2016 | 62 (51) | tropical | |

For the IRS-based retrievals at SGP, MAO, and SMT, cloud base height estimates from collocated ceilometer backscatter profiles are used.

     At SGP, a Raman lidar is operated continuously and provides 10-min averages of water vapor mixing ratio profiles with 60 m vertical resolution. We used water vapor profiles which are merged from narrow and wide fields of view (Turner and Goldsmith, 1999; Newsom and Sivaraman, 2018).

**2.2   Retrieval**

The TROPoe retrieval determines the optimal state vector $\boldsymbol{X}$ which satisfies both the observations and the prior. In our study, the state vector consists of temperature and water vapor profiles as well as liquid water path (LWP). Starting with the mean prior $\boldsymbol{X}_a$ as a first guess of the state vector, a forward model $F$ is used to compute pseudo-observations, which are then compared



to the actual observations. The retrieval iterates until the differences between the pseudo-observations and the observations are small within the uncertainty of the measurements. The state vector at the $n+1$ iteration is computed as:

$$\boldsymbol{X}_{n+1} = X_a + (\gamma \boldsymbol{S}_a^{-1} + \boldsymbol{K}_n^T \boldsymbol{S}_\epsilon^{-1} \boldsymbol{K}_n)^{-1} \boldsymbol{K}_n^T \boldsymbol{S}_\epsilon^{-1} (\boldsymbol{Y} - F(\boldsymbol{X}_n) + \boldsymbol{K}_n (\boldsymbol{X}_n - \boldsymbol{X}_a)) \tag{1}$$

where $\boldsymbol{K}$ is the Jacobian of $F$, $\boldsymbol{S}_a$ is the covariance matrix of the prior, $\boldsymbol{Y}$ is the observation vector, and $\boldsymbol{S}_\epsilon$ denotes the error covariance matrix of the input. Ideally, $\boldsymbol{S}_\epsilon$ includes the error covariance matrix of the observations and forward model. Since the forward model uncertainty is assumed to be zero in the current TROPoe framework, $\boldsymbol{S}_\epsilon$ equals the error covariance matrix of the observations $\boldsymbol{S_Y}$. The scalar $\gamma$ is used to stabilize the retrieval when $n$ is small. It is a function of iteration number and cycles through a fixed sequence of integer values ranging from 1000 to 1. It decreases to unity for larger $n$ and is used to change the relative weight between the prior information and the observation, where $\gamma > 1$ corresponds to less information from the observations relative to the prior (more details are provided in Turner and Löhnert, 2014). As the forward model, we use the Line-By-Line Radiative Transfer Model LBLRTM (Clough et al., 2005) for the IRS-based retrievals and the Monochromatic Radiative Transfer Model MonoRTM (Clough et al., 2005) for the MWR-based retrievals.

The climatological prior is computed specifically for each site and each month using a large number (>1000) of radiosondes profiles. At all sites but SAV, the radiosondes for the prior computation were launched directly at the sites. To compute the prior for SAV, operational radiosondes launched at Abidjan in Ivory Coast (approximately 750 km away) were used. The conditions on individual days may still significantly differ from the monthly mean profile. Hence, we additionally recenter the monthly mean prior profile using the daily average of near-surface water vapor mixing ratio. The temperature profile is recentered by conserving the relative humidity profile. Recentering the monthly prior can help to reduce the retrieval error for both IRS and MWR-based retrievals (Turner and Adler, 2024).

In general, thermodynamic profiles are retrieved on 55 vertical levels reaching from the surface up to 17 km, with the distance between levels starting at 10 m and increasing geometrically with height. Due to the very dry conditions at SMT, the number of molecules in a geometrically thin layer was not sufficient to get enough signal in the microwave range and we had to increase the distance between levels and reduce the number of vertical levels to 33 for the MWR-based retrievals.

TROPoe provides a number of output variables which allow one to characterize the information content of the solution and to distinguish between solutions with good and dubious quality. The retrieval outputs two matrices, the averaging kernel, $\boldsymbol{Akernel}$, and the posterior covariance matrix, $\boldsymbol{S}_{op}$ (Turner and Löhnert, 2014). The square root of $\boldsymbol{S}_{op}$ specifies the 1-$\sigma$ uncertainty of the temperature and water vapor mixing ratio profiles $[\boldsymbol{\sigma}_T, \boldsymbol{\sigma}_{WVMR}]^T$. The diagonal components of the $\boldsymbol{Akernel}$ provide the Degree of Freedom for Signal (DFS), which is a measure of number of independent pieces of information from the observations used in the solution for each height. The cumulative DFS (cDFS) is computed as the trace (i.e., the sum of the diagonal components) of the $\boldsymbol{Akernel}$. The rows of the $\boldsymbol{Akernel}$ give a measure of the smoothing functions of the retrieval as a function of height (Rodgers, 2000) and can be applied to thermodynamic profiles from radiosondes or Raman lidar with higher vertical resolution to minimize the vertical representativeness error when comparing TROPoe profiles to such profiles. The $\boldsymbol{Akernel}$ is different for every experiment of TROPoe, because different observational inputs are used in these experiments,





which results in different **_Akernel_**-smoothed profiles of radiosondes for each configuration. This is why we do not use the **_Akernel_** smoothed radiosonde profiles when evaluating the TROPoe errors to assure the same reference profile is used for each experiment. The radiosonde profiles are instead interpolated to the vertical grid of the retrieved profiles.

To distinguish between profiles with good vs. dubious quality, we used two variables to filter the profiles: $\gamma$ from Eq. 1 and the root mean square error between IRS radiances and MWR brightness temperatures in the observations and the forward calculation:

$$RMSR = \sqrt{\frac{1}{M} \sum_{i=1}^{M} \left( \frac{Y_i - F(X_n)_i}{\sigma_{Y_i}} \right)^2} \qquad (2)$$

with $\sigma_{Y_i}$ being the 1-$\sigma$ uncertainty of the radiance (or brightness temperature) observations and $M$ being the length of the
observation vector. Large RMSR values indicate a large discrepancy between the solution and the observations even though the retrieval found a solution. To filter out suspicious profile we require $\gamma = 1$ and $RMSR < 5$.

IRS-based retrievals have little to no information content above cloud base. This is why we do not use any thermodynamic data above cloud base when LWP > 8 g m$^{-2}$ for the analysis of profiles and exclude any cloudy profiles in our statistical analysis. This limitation does not apply to MWR-based retrievals, due to the transparency of clouds in the microwave range.
We still exclude profiles when LWP > 200 g m$^{-2}$ to screen out cases when clouds are raining for which our assumption of being in the Rayleigh scattering regime is not correct.

## 3    TROPoe experiments

We ran TROPoe with four different configurations for IRS data and with two different configurations for MWR data. The experiments for IRS data build on one another, with the last one including all the changes made in the previous experiments.
The experiments are overviewed in Tab. 2 and will be described in detail in the following subsections. Figure 3 illustrates the impact that the different experiments have on the retrieved thermodynamic profiles using time-height cross sections of water vapor mixing for the IRS-based retrieval at SGP on a relatively moist day in August as an example. Figure 4 shows time series of some diagnostics provided by the retrieval to better understand the impacts.

### 3.1    CTRL

The settings in CTRL are currently used in routine applications of TROPoe. Thermodynamic profiles and LWP were retrieved every 10 min from the instantaneous radiance (or brightness temperature) measurements. In addition to the measurements of IRS and MWR, respectively, ($\boldsymbol{Y}_{IRS/MWR}$), collocated surface measurements of temperature and humidity ($\boldsymbol{Y}_{Met}$) as well as temperature and humidity profiles above 4 km from radiosondes ($\boldsymbol{Y}_{Raso}$) are used as temporally resolved input data in the observation vector in Eq. 1:

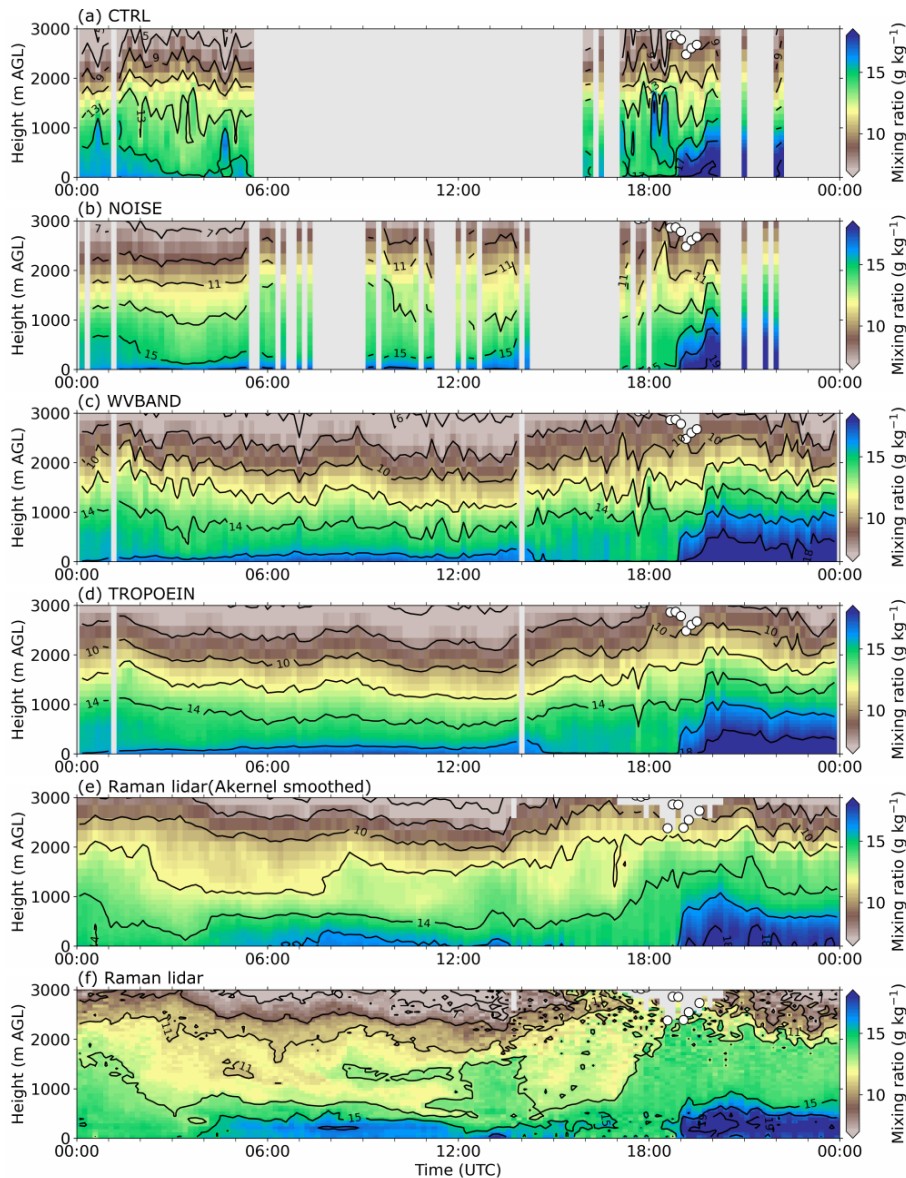

**Figure 3.** Time-height cross section of water vapor mixing ratio retrieved with the IRS-based TROPoe experiments (a) CTRL, (b) NOISE, (c) WVBAND, and (d) TROPOEIN and observed by the Raman lidar (e) when smoothed with the of TROPOEIN and (f) with original vertical resolution at SGP on August 7 2019. The white markers indicate cloud base height from the TROPoe output (a-d) and as detected by the Raman lidar (e-f).

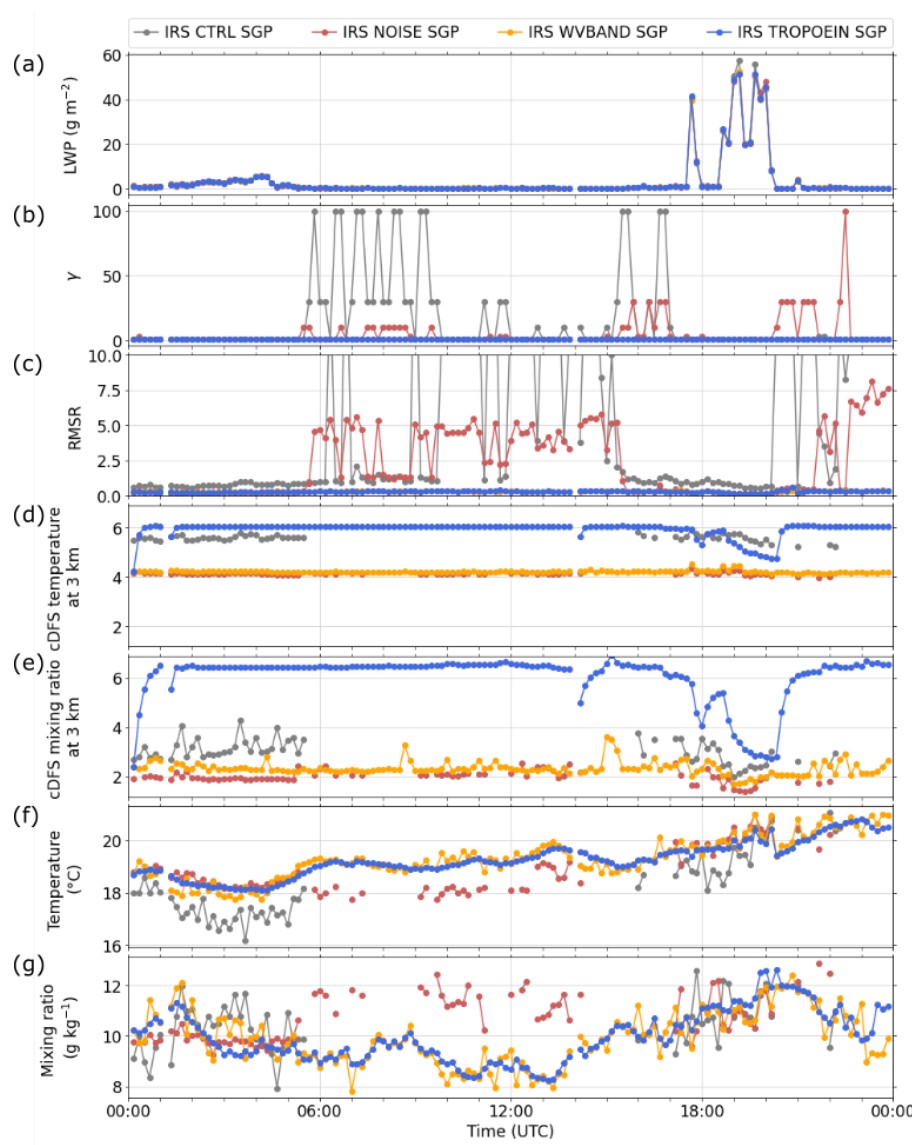

**Figure 4.** Time series of (a) liquid water path LWP, (b) parameter $\gamma$ (Eq. 1), (c) root mean square error of radiance RMSR (Eq. 2 ), (d) cumulative degree of freedom cDFS of water vapor mixing ratio at 3 km, (e) cDFS of temperature at 3 km, (f) temperature at 2 km, and (g) water vapor mixing ratio at 2 km in different IRS-based TROPoe experiments at SGP on August 7 2019.



**Table 2.** Overview of TROPoe experiments for IRS and MWR-based retrievals.

| Configuration | Instrument | Minimum IRS noise level | Additional IRS water vapor band | Previous thermody-namic profile as input |
|---|---|---|---|---|
| CTRL | IRS | | | |
| NOISE | IRS | x | | |
| WVBAND | IRS | x | x | |
| TROPOEIN | IRS | x | x | x |
| CTRL | MWR | | | |
| TROPOEIN | MWR | | | x |

$$\mathbf{Y} = \begin{bmatrix} \mathbf{Y}_{IRS/MWR} \\ \mathbf{Y}_{MET} \\ \mathbf{Y}_{Raso} \end{bmatrix} \qquad\qquad (3)$$

Using IRS radiances, CTRL fails to provide a valid profile for more than 60 % of the example day in Fig. 3 leading to large data gaps and making the analysis of diurnal cycles impossible. The reason for the large number of invalid profiles are high $\gamma$ and high RMSR values exceeding their respective thresholds (Fig. 4b,c).

### 3.2 NOISE

The second experiment only applies to IRS-based retrievals and uses a specified minimum noise level for the radiances (NOISE). As explained in Sect. 1, the radiance noise has to be large enough to compensate for the missing forward model error in the current TROPoe framework. Figure 5 illustrates noise levels at the different sites, with the lowest noise levels at MAO and SGP in August and the highest noise level at SMT. The purple line indicates the default minimum noise level which we propose to use in TROPoe. This default minimum noise level is a tradeoff between availability of valid profiles, information 210 content, and temperature and humidity errors when comparing the TROPoe retrievals to radiosondes launched at SGP during the whole year of 2019 (Appendix A). We will show in Sect. 4 that this experiment has the largest impact on the solution at MAO and SGP, since the minimum noise level is usually higher than the IRS noise at these sites. It has little impact at SMT where the IRS noise is already higher than the minimum noise in most parts of the spectrum.

On the example day in Fig. 3, the number of valid profiles is increased for NOISE compared to CTRL, because $\gamma$ and RMSR 215 (Fig. 4b,c) are lower. However, NOISE still failed to provide valid retrieved profiles on about 45 % of the day.

### 3.3 WVBAND

The third experiment also only applies to IRS-based retrievals and includes an additional water vapor band between 793 and 804 cm$^{-1}$ (Fig. 1) in addition to the default minimum noise level. This additional water vapor band is only used when the





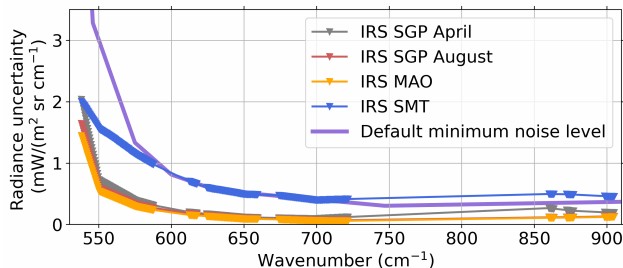

**Figure 5.** Typical spectral radiance uncertainty of the IRS at SGP, MAO, and SMT. The purple line indicates the default minimum noise level used in NOISE, WVBAND, and TROPOEIN.

environment is moist. We found that in dry environments with low water vapor values, some $CO_2$ absorption bands in this
range become dominant and degrade the retrieval performance. We address this in TROPoe by artificially inflating the noise in this band as a function of near-surface water vapor mixing ratio with no inflation for values larger than 12 g kg$^{-1}$ and an inflation factor of 20 for water vapor mixing ratio values of less than 8 g kg$^{-1}$ which basically shuts off the use of any information in this band when the water vapor mixing ratio is low. In between, the inflation factor changes linearly.

Adding the additional water vapor band helps tremendously in the example shown in Fig. 3. The retrieval now provides
valid solutions 100 % of the time (the two gaps at 1:00 and 14:00 UTC are due to a closed hatch) with $\gamma = 1$ and $RMSR < 5$ throughout the day (Fig. 4b,c).

### 3.4 TROPOEIN

The fourth experiment TROPOEIN is applied to both the IRS and MWR-based retrievals (Tab. 2). In general, the TROPoe retrievals are well capable in resolving mesoscale changes that occur on time scales of multiple hours. In the example in
Fig. 3e,f, the Raman lidar detects an increase in water vapor mixing ratio in the lowest 1000 m after around 19 UTC. The timing and magnitude of this increase is well captured in WVBAND (Fig. 3c). Because the vertical resolution of the Raman lidar is higher than the TROPoe profiles, we use the rows of the ***Akernel*** to compute smoothed Raman lidar profiles for comparison to the retrieval.

On the shorter time scales, the TROPoe retrieval looks noisier than the ***Akernel*** smoothed Raman lidar profiles (Fig. 3c,e).
To better visualize this, we applied a high-pass and low-pass filter to the time series with a cut-off time of around 3 h. We tested cut-off times between 3 and 9 h and found little sensitivity on the following results to the chosen value. The high-pass filtered data are shown in color and the low-pass filtered data as contours in Fig. 6. As expected, the low-pass filtered data are similar for all TROPoe experiments and the Raman lidar. The high-pass filtered mixing ratio values of CTRL, NOISE, and WVBAND, however, have higher magnitudes and more variability from one time-stamp to the other compared to the Raman
lidar. The purpose of the TROPOEIN configuration is to improve the agreement between the high-pass filtered TROPoe and Raman lidar data.

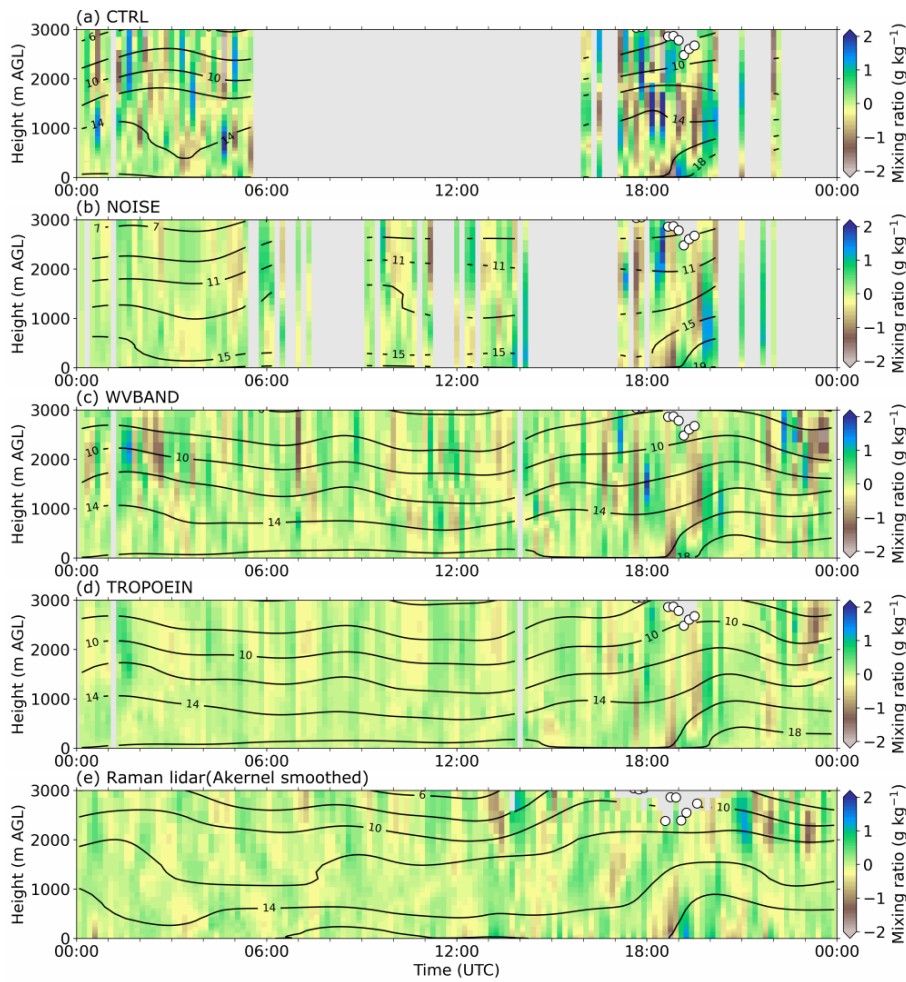

**Figure 6.** Time-height cross section of high-pass (color) and low-pass (contours) water vapor mixing ratio with a cut-off time of approximately 3 h retrieved retrieved with the IRS-based TROPoe experiments (a) CTRL, (b) NOISE, (c) WVBAND, and (d) TROPOEIN and (e) observed by the Raman lidar when smoothed with the ***Akernel*** of TROPOEIN at SGP on August 7 2019.



The idea for TROPOEIN is based on the assumption that the atmosphere is autocorrelated for some period of time. We inspected the time series of water vapor mixing ratio profiles measured by the Raman lidar at the SGP site with 10 min temporal resolution at different heights for the whole year of 2019 and computed monthly autocorrelations. While the magnitude of the

decrease in autocorrelation varies from month to month, the autocorrelation is always more than 0.8 for a lag of 2 h indicating a high degree of correlation over this time period. To account for this temporal autocorrelation, we add a previous valid TROPoe profile of temperature and humidity ($\boldsymbol{Y}_{TROPoe}$) as input to the observation vector:

$$\boldsymbol{Y} = \begin{bmatrix} \boldsymbol{Y}_{IRS/MWR} \\ \boldsymbol{Y}_{MET} \\ \boldsymbol{Y}_{Raso} \\ \boldsymbol{Y}_{TROPoe} \end{bmatrix} \tag{4}$$

with the observational error covariance matrix

$$\boldsymbol{S_Y} = \begin{bmatrix} \boldsymbol{S}_{IRS/MWR} & 0 & 0 & 0 \\ 0 & \boldsymbol{S}_{MET} & 0 & 0 \\ 0 & 0 & \boldsymbol{S}_{Raso} & 0 \\ 0 & 0 & 0 & \boldsymbol{S}_{TROPoe} \end{bmatrix}. \tag{5}$$

The retrieved profiles are thus not independent anymore and we have to assure that we do not suppress any real temporal variations. This is done by inflating the 1-$\sigma$ uncertainty of the previous retrieved profile $[\boldsymbol{\sigma}_T, \boldsymbol{\sigma}_{WVMR}]^T$ before it is used as $\boldsymbol{S}_{TROPoe}$.

Because MWRs and IRSs have the highest information content in the lowest layers and because changes in the boundary

layer typically can happen more rapidly than in the free troposphere, we do not want to constrain the solution in lower layers too much by the previous profile. Thus, we increase the uncertainties $[\boldsymbol{\sigma}_T, \boldsymbol{\sigma}_{WVMR}]^T$ by a height-dependent additive factor for temperature $\boldsymbol{N}_T$ and noise multiplier for water vapor mixing ratio $\boldsymbol{N}_{WVMR}$ (gray lines in Fig. 7). We set $\boldsymbol{N}_T$ to decrease from 3 °C at the surface to 1 °C at either the top of the boundary layer or at 1 km, whatever is higher, and to stay constant above. $\boldsymbol{N}_{WVMR}$ is set to decrease from 5 to 2 at the maximum of 1 km or the boundary layer top and also stays constant above.

The boundary layer height was retrieved from potential temperature profiles retrieved with TROPoe using the parcel method (e.g., Duncan Jr. et al., 2022). To account for the decrease in autocorrelation with time, $\boldsymbol{N}_T$ and $\boldsymbol{N}_{WVMR}$ are increased by a time-dependent factor

$$fac_{\Delta t} = \sqrt{1 + \frac{\Delta t - t_{res}}{t_{res}}} \tag{6}$$

with $\Delta t$ being the elapsed time in seconds between $\boldsymbol{Y}_{TROPoe}$ and the current time being processed, and $t_{res}$ being the

temporal resolution of the retrieval in seconds. This means that instead of setting a fixed limit on how far back in time profiles are used, the uncertainty is increased with time so that the impact from the previous profiles gradually fades when $\Delta t$ increases. Figure 8 visualizes how $fac_{\Delta t}$ increases with $\Delta t$.

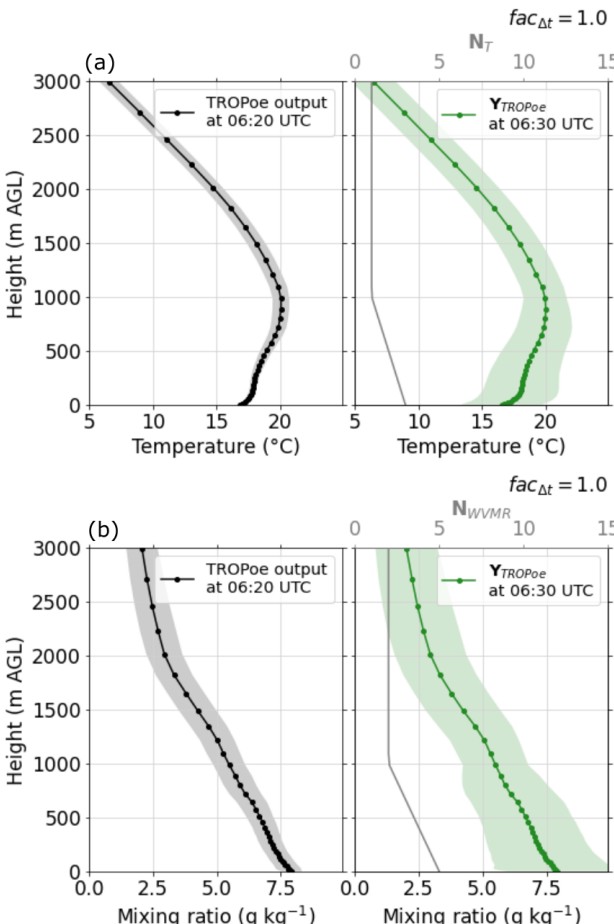

**Figure 7.** Demonstration of noise inflation for the previous retrieved profile when it gets used as additional input in TROPOEIN, (a) for temperature and (b) for water vapor mixing ratio. The black lines are the retrieved profiles at 6:20 UTC at SGP on April 21 2019 with the gray shading indicating $\sigma_T$ and $\sigma_{WVMR}$, respectively. The green lines are the input profiles used in the observation vector $\boldsymbol{Y}_{TROPoe}$ for the retrieval at 6:30 UTC with the green shading indicating the corresponding uncertainty $\boldsymbol{S}_{TROPoe}$ (Eq. 7). The gray lines indicate the (a) additive factor $\boldsymbol{N}_T$ in deg C and (b) multiplier $\boldsymbol{N}_{WVMR}$ used for inflating the $\sigma_T$ and $\sigma_{WVMR}$ profiles.



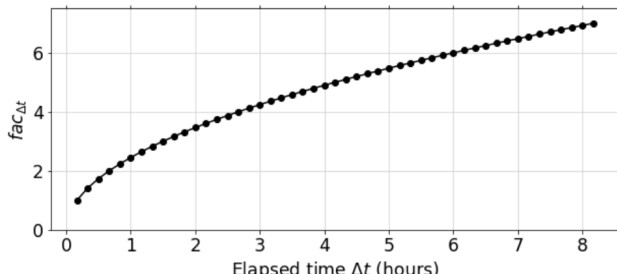

**Figure 8.** Increase of the time-dependent factor $fac_{\Delta t}$ (Eq. 6) with the time elapsed between the last valid retrieved profile and the time currently being processed with TROPoe $\Delta t$ for the TROPOEIN experiment.

$\boldsymbol{S}_{TROPoe}$ is hence computed as

$$\boldsymbol{S}_{TROPoe} = \begin{bmatrix} \boldsymbol{S}_{TROPoe,T} \\ \boldsymbol{S}_{TROPoe,WVMR} \end{bmatrix} = \begin{bmatrix} (fac_{\Delta t} \cdot \boldsymbol{N}_T) + \boldsymbol{\sigma}_T \\ (fac_{\Delta t} \cdot \boldsymbol{N}_{WVMR}) \cdot \boldsymbol{\sigma}_{WVMR} \end{bmatrix} \tag{7}$$

Figure 7 illustrates the uncertainty inflation for an IRS-based retrieval at the SGP site. The black lines show the retrieved water vapor and temperature profile at 06:20 UTC. The black shading indicates the respective 1-$\sigma$ uncertainty. This profile is then used as $\boldsymbol{Y}_{TROPoe}$ for the retrieval at 06:30 UTC with the inflated uncertainty $\boldsymbol{S}_{TROPoe}$ (green shading). The impact of the uncertainty inflation is highest close to the ground, because of the height-dependent additive factor and multiplier (grey lines).

A good way to understand the impact of the uncertainty inflation with time on the solution is by looking at cDFS at 3 km using the IRS-based retrieval as an example (Fig. 4d,e). Since typically we start the processing of each day independently, $\boldsymbol{Y}_{TROPoe}$ is unavailable at 00:00 UTC and cDFS is the same as for WVBAND. cDFS increases for the next couple of time stamps and levels after around 01:00 UTC. Because of the missing profiles at around 01:00 and 14:00 UTC, cDFS after the data gap is lower because $\boldsymbol{S}_{TROPoe}$ is increased according to Eq. 7. After 17:00 UTC, clouds with cloud bases around 2500 m

AGL (dots in Fig. 3) and LWP of up to 50 g m$^{-2}$ (Fig. 4a) are present. Profiles with a LWP > 8 g m$^{-2}$ are not used as input due to the limitation of the IRS in the presence of clouds, which is why cDFS gradually decreases with time during the cloudy period when $\Delta t$ increases. When $\Delta t$ nears around 2 h, hardly any impact of the last good profile was visible anymore and cDFS in TROPOEIN has approached the values in WVBAND. The factor to increase the uncertainty with time (Eq. 6) is chosen so that the impact of the previous profiles diminishes after around 2 h, taking into account the decrease in autocorrelation in time

which we find for the Raman lidar data. After the clouds clear, cDFS increases again because valid profiles closer in time are again used as input.

    The smoothing impact TROPOEIN has on the water vapor time series is directly visible in Figs. 3d, 4f,g, and 6d. The temporal variability in the time series is much reduced compared to WVBAND (Figs. 3c, 6c) and the magnitude of the high-pass filtered values is now very similar to the one of the Raman lidar (Fig. 6e).



To quantify the agreement between the high-pass filtered TROPoe and Raman lidar data, we use two measures: the uncorrelated random noise in TROPoe and Raman lidar time series and the correlation between the TROPoe and the Raman lidar time series. Information on the uncorrelated random noise of a time series can be obtained from its autocorrelation. The uncorrelated random noise $\Delta M$ of a time series $\boldsymbol{x}$ with length $N$ can be estimated from the autocovariance function at lag $\tau$:

$$M(\tau) = \frac{1}{N} \sum_{i=1}^{N-\tau} (x_i)(x_{i+\tau})$$

as the difference between the first two lags (Lenschow et al., 2000)

$$\Delta M = M(0) - M(1).$$

The autocorrelation function is computed as $R(\tau) = M(\tau)/M(0)$ and is 1 at lag 0. The uncorrelated random noise thus relates to the autocorrelation function at lag 1 as follows:

$$R(1) = 1 - \frac{\Delta M}{M(0)}.$$

Before computing the autocovariance function, we removed all missing data in the high-pass filtered time series and simply stitched the data together only using time stamps when all four TROPoe experiments and the Raman lidar provided valid data. Another method to estimate the autocovariance of time series with gaps is based on the Lomb-Scargle periodogram (VanderPlas, 2018). We tested both methods and found the same relationship for uncorrelated noise, which is why we choose the simpler method of using the autocovariance for our analysis.

The Pearson correlation coefficient $r$ between time series of TROPoe data $\boldsymbol{x}_{TROPoe}$ and of Raman lidar data $\boldsymbol{x}_{Raman}$ is computed as

$$r = \frac{\sum_{i=1}^{N} (x_{TROPoe,i} - \overline{\boldsymbol{x}_{TROPoe}})(x_{Raman,i} - \overline{\boldsymbol{x}_{Raman}})}{\sqrt{\sum_{i=1}^{N} (x_{TROPoe,i} - \overline{\boldsymbol{x}_{TROPoe}})} \sqrt{\sum_{i=1}^{N} (x_{Raman,i} - \overline{\boldsymbol{x}_{Raman}})}} \tag{8}$$

with the overbar indicating temporal average.

## 4 Sensitivity of thermodynamic profiles to TROPoe experiments

### 4.1 Solution availability and information content

The example in Fig. 3 illustrates how the availability of valid solutions increases in NOISE and WVBAND compared to CTRL on a day in August at SGP, because of a reduction in $\gamma$ and $RMSR$ values (Fig. 4b,c). Figure 9 now shows how the solution availability changes at the individual sites for the month-long periods. The fraction of valid retrievals in CTRL ranges from just above 50 % at SGP in August to nearly 100 % at SMT (Fig. 9a). Using the default minimum noise level (NOISE), increases





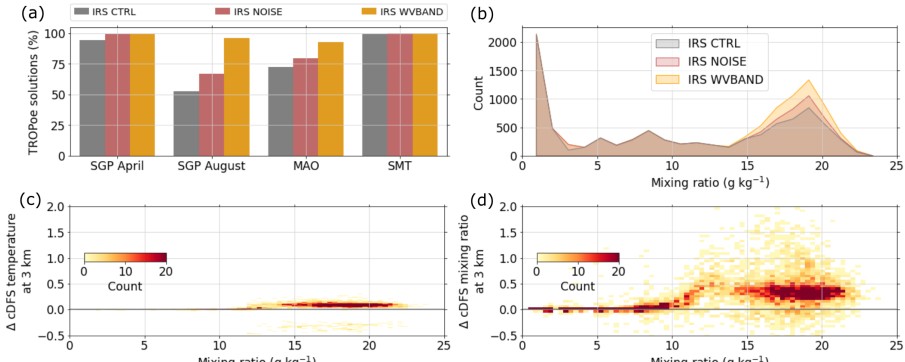

**Figure 9.** Percentage of valid TROPoe solutions for the month-long periods at SGP, MAO and SMT, (b) solution availability as a function of near-surface water vapor mixing ratio, (c) difference in cDFS of temperature at 3 km as a function of near-surface water vapor mixing ratio, and (d) difference in cDFS of water vapor mixing ratio at 3 km as a function of near-surface water vapor mixing ratio. In (b)-(d) all sites and months are included.

the number of solutions at SGP and MAO and is now higher than 65 % at all sites. As expected, no change is visible at SMT due to the already high radiance noise level at this site (Fig. 5).

Adding the additional water vapor band (WVBAND) further increases the profile availability at SGP in August and MAO to more than 92 %. The different impact of WVBAND is linked to the environmental moisture at the individual locations. The near-surface water vapor mixing ratio at SMT and SGP in April is mostly below the threshold of 12 g kg$^{-1}$ (Fig. 2) and

thus in a range where little to no information from the additional band is used in the retrieval (Sect. 3.3). On the other hand, the near-surface water vapor mixing ratio at SGP in August and MAO is usually higher than the threshold and the additional water vapor bands are used in the retrieval. As expected, WVBAND only has a positive impact on solution availability when near-surface mixing ratio is higher than the threshold (Fig. 9b). Importantly, it also does not reduce solution availability in dry environments which means that the inflation of noise in the additional water vapor band as a function of near-surface mixing

ratio works. By including the additional water vapor band in the retrieval, we expect to increase the number of independent pieces of information. The difference in cDFS of water vapor mixing ratio at 3 km between WVBAND and NOISE over near-surface water vapor mixing confirms an increase by around 0.4 for moist environments when adding the additional water vapor band (Fig. 9d). The positive impact on cDFS of temperature is less pronounced, but still present when near-surface water vapor mixing ratio exceeds the threshold of 12 g kg$^{-1}$ (Fig. 9c).

## 4.2   Temporal consistency

### 4.2.1   TROPOEIN for IRS-based retrievals

The example in Figs. 3 and 6 indicates that TROPOEIN smooths the time series of retrieved profiles and thus decreases random uncorrelated noise. To provide a more quantitative and general analysis, we now present profiles of autocorrelation at lag 1,



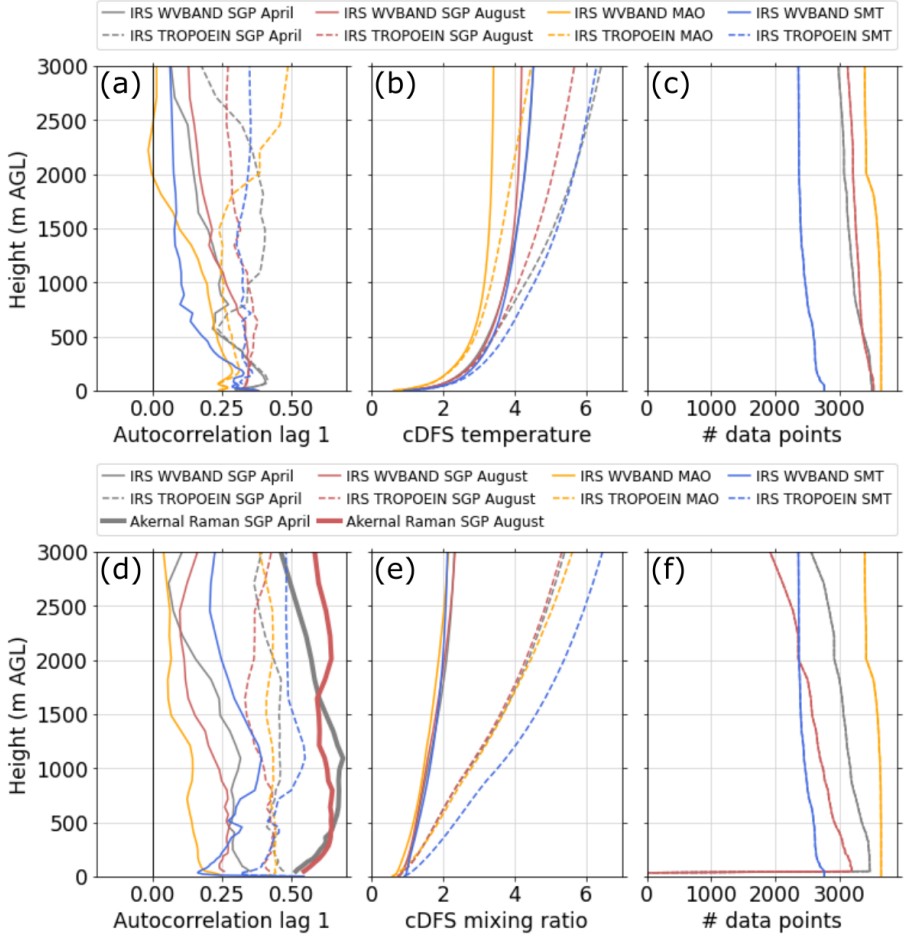

**Figure 10.** Profiles of autocorrelation at lag 1 for high-pass filtered time series of (a) temperature and (d) water vapor mixing ratio, mean cumulative degree of freedom (cDFS) of (b) temperature and (e) water vapor mixing ratio, and available data points for (c) temperature and (f) water vapor mixing ratio profiles for IRS-based retrievals at SGP, MAO, and SMT. In (d), autocorrelation for high-pass filtered *Akernel* smoothed water vapor mixing ratio profiles measured by Raman lidar are added.

as an indicator for uncorrelated random noise, for high-pass filtered temperature and water vapor mixing ratio profiles for

IRS-based retrievals using the month-long periods at the three sites (Fig. 10). We compare WVBAND and TROPOEIN which have identical configurations except for using the previous thermodynamic profile as input in TROPOEIN. The profiles at the individual sites are time-matched; this means that only time stamps are used for which both TROPoe runs have valid data. The number of data points decreases with height since data above cloud base are excluded from the analysis (Fig. 10c,f). For water vapor mixing ratio, the Raman lidar availability is additionally considered, which results in a different number of available data

points for temperature and water vapor profiles at SGP.





Autocorrelation in WVBAND is largely less than 0.25 at all sites (solid lines in Fig. 10a,d), indicating relatively low signal-to-noise ratio for true atmospheric variability. For both temperature and water vapor mixing ratio, autocorrelation decreases with height which is consistent with an increase in noise as the DFS decreases (i.e., cDFS becomes nearly constant with height, Fig. 10b,e). TROPOEIN increases the autocorrelation at all sites and the resulting profiles are roughly constant with height

with values between around 0.3 and 0.45 for both temperature and water vapor (dashed lines in Fig. 10a,d). This indicates that independent of the very different atmospheric conditions at the sites (polar vs mid-latitude vs tropical) and independent of the instrument specifics (different radiance noise), the method enhances the temporal consistency in a similar way. Importantly, the constructed height-dependent increase of uncertainty for the previous profiles (Fig. 7) does not show as artefacts, such as jumps, in the autocorrelation profiles in TROPOEIN.

To evaluate how well the TROPOEIN uncorrelated noise agrees with the Raman lidar noise, we also computed the autocorrelation of high-pass filtered profiles of water vapor mixing ratio measured by the Raman lidar at SGP (Fig. 10d). With values of more than 0.6, autocorrelation is still slightly higher than in TROPOEIN. In the lowest few hundred meters, autocorrelation in the Raman lidar data slightly decreases towards the ground, possibly indicative of higher noise at low altitudes due to the use of the Raman lidars' wide field-of-view near the surface (Turner and Goldsmith, 1999).

TROPOEIN adds information to the profiles, visible in the larger cDFS values on the average compared to WVBAND (Fig. 10b,e). The increase in cDFS is more pronounced for water vapor mixing ratio profiles, which have less signal compared to temperature profiles in WVBAND. Histograms of the cDFS difference between TROPOEIN and WVBAND for individual profiles confirms that cDFS at 3 km is consistently higher in TROPOEIN (Fig. 11a,b). cDFS of water vapor mixing ratio in TROPOEIN exceeds the cDFS in WVBAND by 3 to 5 in (Fig. 11b), while the increase for temperature is mostly between 1

and 2.5 (Fig. 11a).

After we have shown that TROPOEIN decreases uncorrelated random noise in the high-pass filtered retrieved profiles and increases the number of independent pieces of information in the solution, we now investigate how well the retrieval captures temporal atmospheric changes on short time scales. To this purpose, we compute profiles of the Pearson correlation coefficient between the TROPoe and Raman lidar high-pass filtered time series at SGP (Eq. 8), assuming that the Raman lidar represents

the truth (Fig. 12). For both months, the correlation is increased for TROPOEIN at all heights reaching values between 0.5 and 0.6 between approximately 500 and 2000 m, indicating a moderate correlation between the TROPOEIN and Raman lidar mixing ratio profiles. The lower correlation below and above this layer could also be related to higher Raman lidar noise or less valid data points in these levels (Fig. 10d,f).

As previously stated, $S_{TROPoe}$ has to be large enough to not suppress real temporal mesoscale variability. That this is the

case is well visible in the example in Fig. 13 when an elevated layer of moist air was advected over the site after around 09:00 UTC as detected by the Raman lidar. Such elevated moist layers are especially challenging for the retrieval since vertical resolution decreases with height. The retrieval does a remarkable job in resolving this moist layer and captures not only the timing but also the altitude and magnitude of the moisture values. While small-scalle variability is reduced in TROPOEIN (Fig. 13c) compared to WVBAND (Fig. 13b), the mesoscale changes in moisture are represented just as well as in WVBAND,





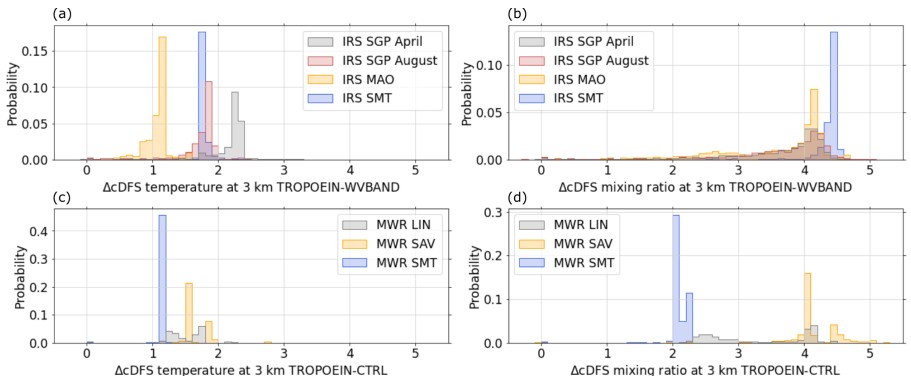

**Figure 11.** Probability of the differences in cDFS for temperature and water vapor mixing ratio (a,b) between TROPOEIN and WVBAND (IRS-based) and (c,d) between TROPOEIN and CTRL (MWR-based). Note the different y-axis ranges.

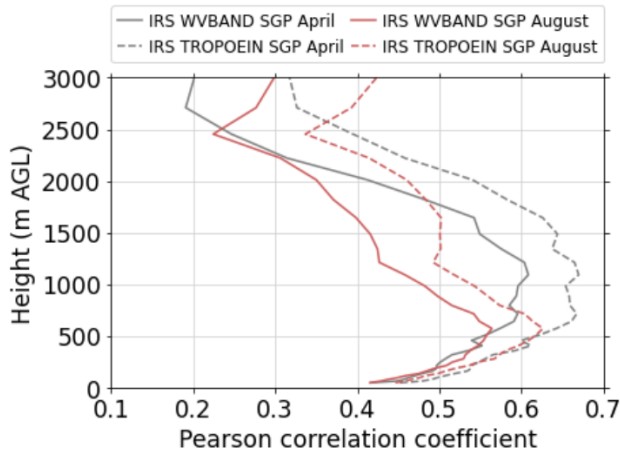

**Figure 12.** Profiles of Pearson correlation coefficient computed between high-pass filtered water vapor mixing ratio time series for the IRS-based TROPoe retrievals WVBAND and TROPOEIN and the Raman lidar at SGP.



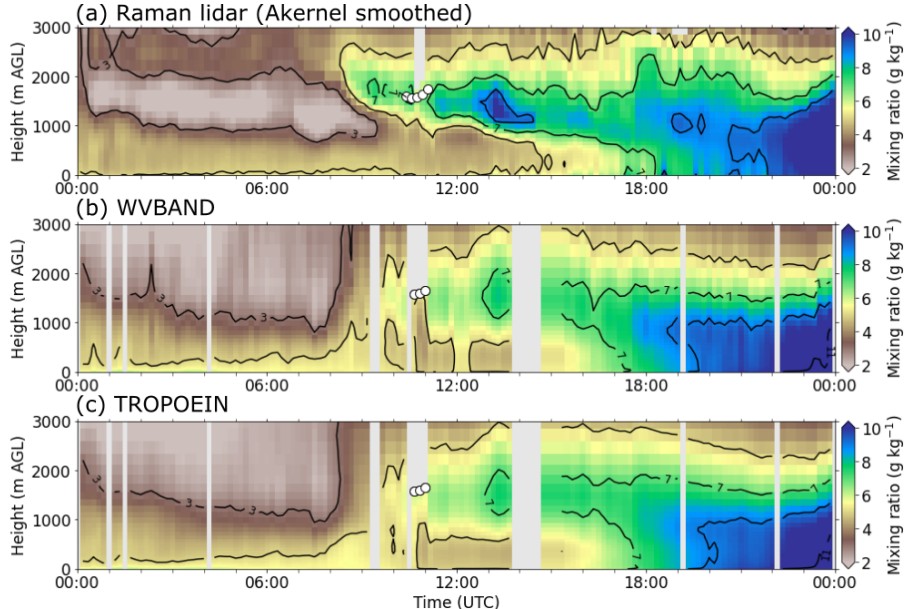

**Figure 13.** Time-height cross section of water vapor mixing ratio from (a) Raman lidar (*Akernel* smoothed), (b) WVBAND, and (c) TROPOEIN on April 28 2019 at SGP. The white markers indicate cloud base height as detected by the Raman lidar (a) and from the TROPoe output (b,c).

which is critical if these observations are being used to initialize numerical weather prediction models (e.g., Coniglio et al., 2019; Hu et al., 2019; Degelia et al., 2020).

### 4.2.2 TROPOE for MWR-based retrievals

We also ran the CTRL and TROPOEIN experiments based on MWR brightness temperatures (Tab. 2). Profiles of autocorrelation at lag 1 for high-pass filtered temperature and humidity time series for MWR-based retrievals are shown in Fig. 14. Unlike
for the IRS-based retrievals, the number of data is constant with height, because data above cloud base were not excluded as clouds are much more transparent at microwave frequencies. The autocorrelation at lag 1 in CTRL was mostly less than 0.3 at all sites and all heights for both temperature and water vapor mixing ratio, indicating high uncorrelated random noise (solid linesin Fig. 14a,d). Including the previous profile as input in TROPOEIN increased the autocorrelation to values between 0.3 and 0.6 (dashed lines). The impact of TROPOEIN was often lower close to the ground, which is consistent with the inflation
of noise in the input profiles towards the surface and the higher information content at low levels, especially for temperature (Fig. 14b,e). Like for the IRS-based retrievals, increases in cDFS are higher for water vapor mixing ratio than for temperature. The number of independent pieces of information at 3 km increased by 2 to 5 for mixing ratio (Fig. 11d) and by 1 to 2 for temperature in TORPOEIN compared to CTRL (Fig. 11c). Overall, the TROPOEIN impacts both IRS and MWR-based retrievals in a similar way and leads to a reduction in uncorrelated random noise and an increase in information content.

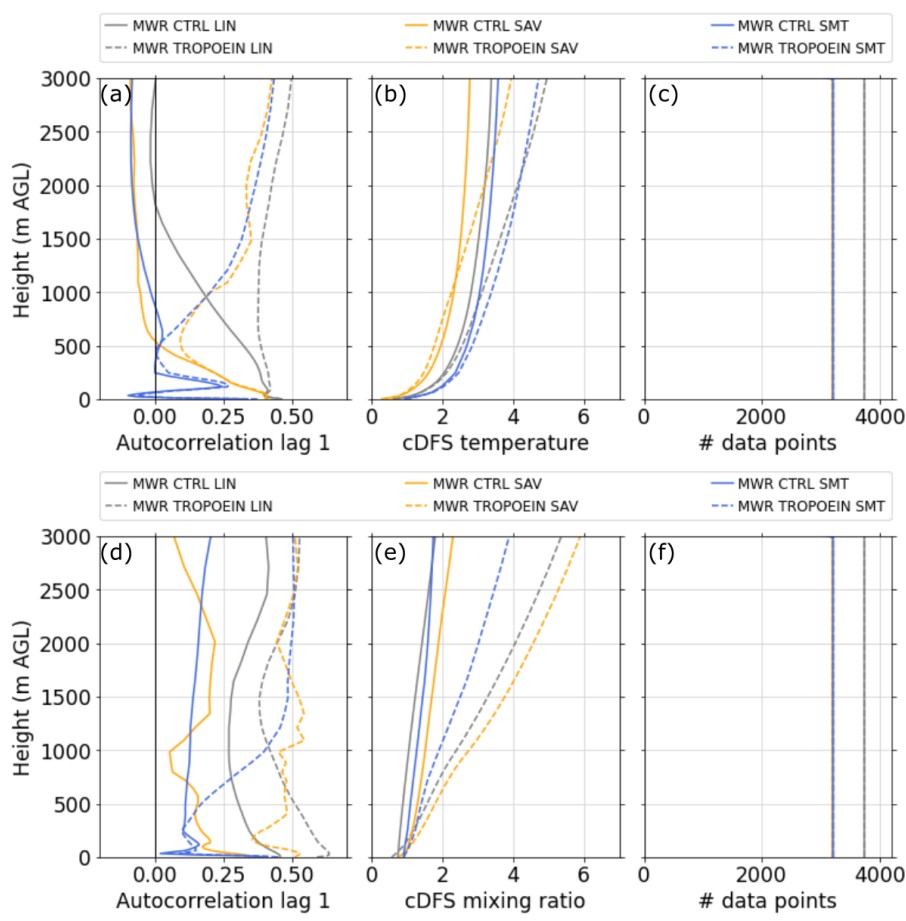

**Figure 14.** Profiles of autocorrelation at lag 1 for high-pass filtered time series of (a) temperature and (d) water vapor mixing ratio, of mean cumulative degree of freedom (cDFS) of (b) temperature and (e) water vapor mixing ratio, and available data points for (c) temperature and (f) water vapor mixing ratio profiles for MWR-based retrievals at LIN, SAV, and SMT. Note that the number of data points at SAV and SMT are nearly identical.



## 4.3   Comparison to radiosonde profiles

In the previous sections, we showed how the modified configurations improve solution availability and temporal consistency. In this section, we now assess the impact of the different experiments on temperature and humidity errors by comparing the retrieved profiles to radiosonde profiles. We computed the mean absolute error (MAE) for temperature and water vapor as well as the mean relative error (MRE) for water vapor averaged up to 3 km for both the IRS and MWR-based retrievals (Fig. 15). The profiles were interpolated to an equidistant grid before computing the MAE. MRE is calculated as MAE divided by the mean water vapor mixing ratio in the lowest 3 km of the radiosonde profile.

Average MAE for temperature mostly range between 0.5 and 1 °C for both IRS-based and MWR-based retrievals (Fig. 15a,b) and MAE for water vapor mixing ratio range between 0.5 and 1.5 g kg$^{-1}$ (Fig. 15c,d) with the exception of SMT where mixing ratio values are very low (Fig. 2). The relative errors are however largest in this very dry polar environment with mean values ranging between 20 and 30 % (Fig. 15e,f). For the four experiments of the IRS-based retrievals, the MAE progressively decreases from CTRL to NOISE to WVBAND to TROPOEIN for both temperature and water vapor mixing ratio at most sites. Only at SGP in August does the average MAE for temperature increase slightly from CTRL to NOISE. When averaged over all sites, MAE for water vapor mixing ratio improves by 6.3 % for NOISE, 11.2 % for WVBAND, and 11.0 % for TROPOEIN relative to CTRL. MAE for temperature improves by 3.2 %, 3.6 %, and 7.4 %, respectively. Changes in MAE between TROPOEIN and CTRL for MWR-based retrievals are very small. This means that TROPOEIN hardly impacts the error compared to radiosondes. This, in combination with the very positive impact on temporal consistency, is very promising and we are working on including the TROPOEIN option in the TROPoe Docker container to make it available to all users.

## 5   Summary and conclusions

The optimal estimation physical retrieval TROPoe combines radiance observations made by passive ground-based remote sensors like MWR and IRS with thermodynamic profiles from active remote sensors, radiosondes, and numerical weather prediction model output to retrieve thermodynamic profiles in the atmospheric boundary layer with high temporal resolution. In this study, we address specific issues in TROPoe related to improving the availability of valid solutions for different atmospheric conditions and to increasing the temporal consistency of the retrieved profiles. To test our modifications to the code, we ran the retrieval with 10-min temporal resolution for one-month long periods using IRS and MWR measurements at tropical, mid-latitude, and polar sites. The main results are:

1. Adequate characterization of the covariance matrix of the different input components, that is the observations, prior, and forward model, is very important for the retrieval performance. Because it would increase the computational costs of the retrieval substantially, the uncertainty of the forward model is currently assumed to be zero and compensated by an inflated observed radiance uncertainty. We found that for IRS-based retrievals this inflation may not be sufficient in some conditions, leading to an overfitting of the data and thus unrealistic solutions. By implementing a default minimum noise level for IRS radiances, the availability of valid solutions increased from around 50 % to more than 65 % at all sites.





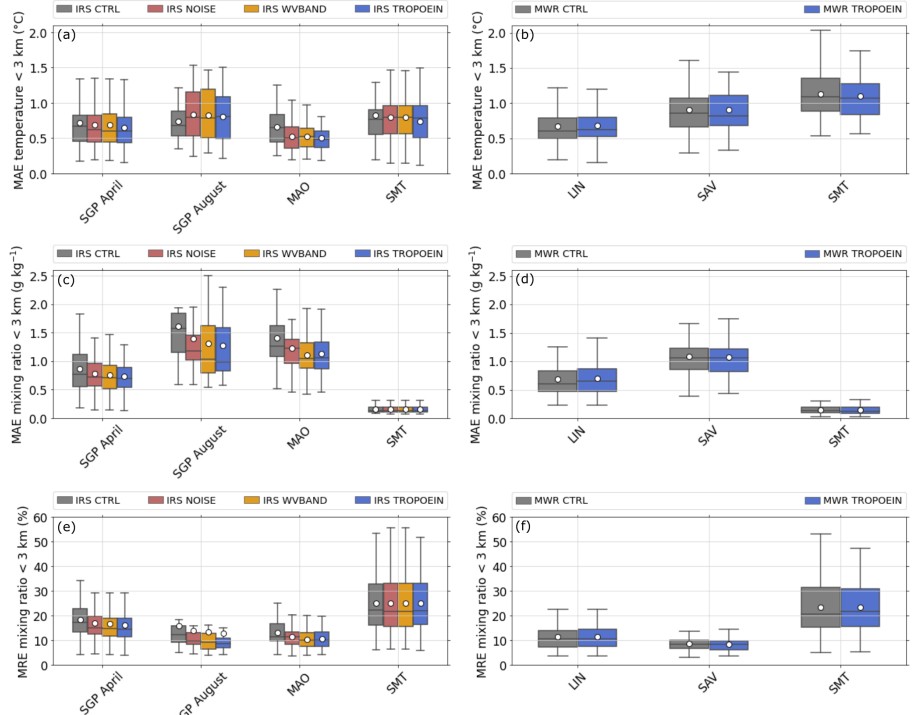

**Figure 15.** Boxplot of mean absolute error MAE averaged below 3 km of (a) temperature and (b) water vapor mixing ratio for IRS-based retrievals and of (c) temperature and (d) water vapor mixing ratio for MWR-based retrievals. Boxplot of mean relative error MRE averaged below 3 km of water vapor mixing ratio for (e) IRS-based retrievals and (f) MWR-based retrievals. MAE and MRE are computed between the radiosonde profiles and the TROPoe profile closest in time within a 30 min window. Only clear-sky profiles when all TROPoe experiments provided valid solutions are considered. The white circles indicate the mean values, boxes show the interquartile range with the median indicated by the horizontal line, and the whiskers extend to the points that lie within 1.5 times the interquartile range of the lower and upper quartiles.

2. In high moisture environments, the traditional infrared spectral bands used for the retrieval of water vapor mixing ratio profiles from ground-based IRS systems (e.g., Turner and Löhnert, 2014; Smith et al., 1999) may be saturated and contain little information content preventing valid retrieval solutions. We showed that by using an additional spectral band 793-804 cm$^{-1}$ in moist conditions (i.e, when the near-surface water vapor mixing > 12 g kg$^{-1}$) the availability of valid solutions increases to more than 92 % at all sites along with an increase in information content. The implementation of the default minimum noise level for IRT radiances and the additional water vapor band, reduces MAE in the lowest 3 km by approximately 11 % for water vapor mixing ratio and by approximately 3.5 % for temperature on average.

3. Time series of the retrieved profiles suffer from uncorrelated random noise and low temporal consistency between subsequent profiles, because every 10-min profile is processed independently without using any information from the previous state of the atmosphere. By including information from a previous retrieved thermodynamic profile as input to the re-



trieval, we take into account the temporal autocorrelation of temperature and humidity in the atmosphere. This method reduces uncorrelated random noise in the retrieved profiles and brings it closer to the noise computed from water vapor mixing ratio profiles measured by Raman lidar. It also increases the number of independent pieces of information. Furthermore, the temporal correlation between high-pass filtered retrieved profiles and the Raman lidar profiles increases, meaning that the retrieval better captures real variations in the atmospheric state on short time scales when including the previous profile.

The first two improvements (default minimum noise level for infrared radiances and additional infrared band for the retrieval of water vapor mixing ratio profiles) are already implemented as default in the current version of TROPoe, while the implementation of the third improvement (using previous TROPoe profile as input) as an option is planned for a future update of TROPoe.

These improvements enhance the value of TROPoe for the study of thermodynamic profiles in the boundary layer at sites in different regions and climates. The higher availability of valid solutions and the increased temporal consistency better allow the analysis of diurnal cycles and temporal tendencies in the boundary layer. This is not only beneficial for case studies to enhance the understanding of physical processes, but also provides better data sets for model evaluation and data assimilation.

*Code and data availability.* The data sets used from the Atmospheric Radiation Measurement (ARM) facilities at Southern Great Plains (SGP) and Manacapuru (MAO) are the following: AERI summary data (Gero et al., 2023b), AERI noise filtered Ch1 data (Zhang, 2023) which are based on Ch1 data (Gero et al., 2023a), surface meteorological data (Kyrouac and Shi, 2023), ceilometer data (Morris et al., 2023), radiosonde data (Keeler and Burk, 2023), and Raman lidar data (Zhang and Newsom, 2023) (SGP only). Data from Summit Station were collected as part of the NSF-funded ICECAPS program (Shupe et al., 2013). At Summit Station (SMT), we used infrared spectrometer data from Walden (2015), microwave radiometer data from Turner and Bennartz (2015), and radiosonde data from Walden and Shupe (2015). Note that these datastreams are also in the ARM data archive, which is where we downloaded them. At Lindenberg (LIN), the MWR measurements (Löhnert et al., 2022) and radiosonde launches (Kirsch et al., 2022) were performed in the framework of the FESSTVaL campaign (Hohenegger et al., 2023). The MWR measurements (Kalthoff, 2016) and radiosonde launches (Lohou, 2016) at Save (SAV) were conducted for the DACCIWA field campaign (Kalthoff et al., 2018). To run the retrieval, we used TROPoe version 0.11 (available from DockerHub as davidturner53/tropoe/v0.11).

## Appendix A: IRS minimum noise level

The uncertainty of the forward model is not included in the current framework of TROPoe. This missing uncertainty has to be compensated by the uncertainty of the observed infrared radiances for IRT-based retrievals. If the instrument specific radiance uncertainty is too low, it is insufficient to compensate the missing forward model uncertainty resulting in overfitting of the data. In these cases, the retrievals may struggle to find a valid solution.

As an intermediate solution before a computational efficient implementation of the forward model uncertainty can be included in TROPoe, we propose a minimum noise level to be used in IRS-based retrievals. To demonstrate the impact of different



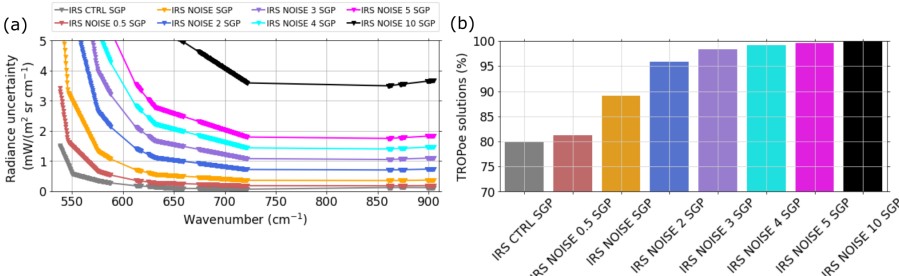

**Figure A1.** (a) Median radiance uncertainty used in the IRS-based retrieval CTRL and NOISE with different default minimum noise levels. (b) Number of valid TROPoe solutions for the experiments with different minimum noise levels (Fig. A1) at the times of the 959 clear-sky radiosonde launches at SGP during the whole year of 2019. NOISE uses the default minimum noise level which we implemented in TROPoe. The numbers 0.5, 2, 3, 4, 5 and 10 indicate the factors by which the noise level was multiplied using NOISE as reference.

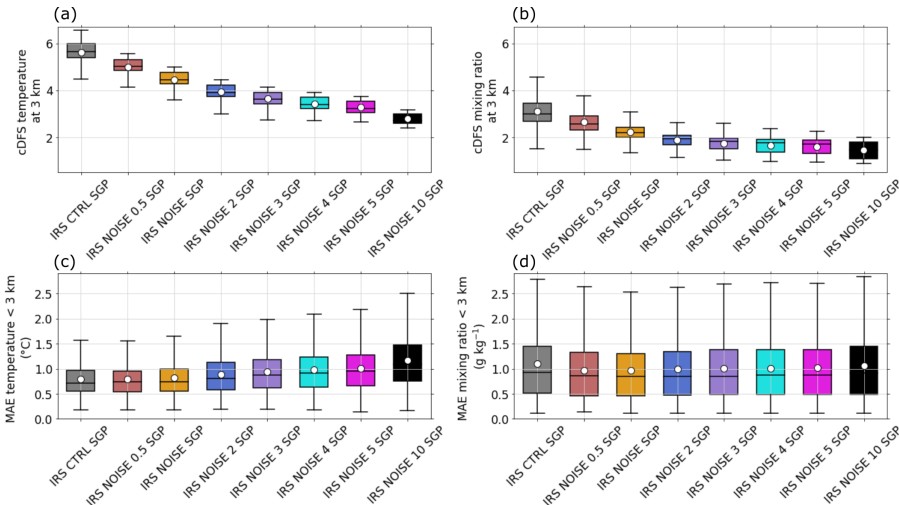

**Figure A2.** Cumulative degree of freedom (cDFS) of (a) temperature and (b) water vapor mixing ratio at 3 km AGL and mean absolute error (MAE) of (c) temperature and (d) water vapor mixing ratio averaged below 3 km AGL when comparing the different TROPoe experiments (Fig. A1) to radiosonde profiles. Only profiles were considered for which all experiments provided valid solutions (around 700).



minimum noise levels on the retrieval solution, we ran TROPoe at the four-times daily radiosonde launch times at SGP for the full year of 2019. The radiance uncertainties for the different experiments are shown in Fig. A1a. The proposed minimum noise level is labeled NOISE. We tested the sensitivity of the solution to experiments with 0.5, 2, 3, 4, 5, and 10 times the noise level in NOISE and also included CTRL for comparison. Importantly, we are assuming this noise is spectrally uncorrelated. Figure A1b shows the percentage of valid solutions for clear-sky retrievals. CTRL provided valid solutions at around 80 % of all profiles, with NOISE 0.5 being only slightly higher. The number of valid solutions increased to nearly 90 % for NOISE and to more than 95 % for NOISE 2 and up. Increasing the radiance noise, decreases the information content of the solution, illustrated by cDFS of temperature and water vapor mixing ratio at 3 km (Fig. A2a,b). When comparing to radiosonde profiles, MAE for temperature was similarly low for CTRL, NOISE 0.5 and NOISE on average, and increased for NOISE 2 and up (Fig. A2c). The average MAE for water vapor was smallest for NOISE (Fig. A2d). The higher number of valid solutions in experiments with high uncertainties of the IRS radiances comes at the cost of lower information content and larger errors compared to radiosonde profiles. This is why we propose NOISE as the default minimum noise level as a tradeoff, however, this value can be changed by the TROPoe user.

*Author contributions.* BA performed the TROPoe experiments, completed the analysis, and prepared the manuscript. DDT helped with the TROPoe experiments and implemented the modfiications in the TROPoe container code. DDT, LB, IVD, JMW, and TM contributed to the discussion and preparation of the manuscript.

*Competing interests.* One of the (co-)authors is a member of the editorial board of Atmospheric Measurement Techniques.

*Acknowledgements.* Funding for this work was provided by the NOAA Physical Sciences Laboratory, the U.S. Department of Energy (DOE) Office of Energy Efficiency and Renewable Energy, Wind Energy Technologies Office, and by the NOAA Atmospheric Science for Renewable Energy (ASRE) program. This research was supported by NOAA cooperative agreement NA22OAR4320151, for the Cooperative Institute for Earth System Research and Data Science (CIESRDS). We specifically thank Martin Kohler from Karlsruhe Institute of Technology for providing the MWR data at Save. The statements, findings, conclusions, and recommendations are those of the author(s) and do not necessarily reflect the views of NOAA or the U.S. Department of Commerce.



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
