# Peer review of "Improving solution availability and temporal consistency of an optimal estimation physical retrieval for ground-based thermodynamic boundary layer profiling"

_EGUsphere, 2024_

## Author Comment (AC1)

**Response to comments of reviewer 1**

We thank the anonymous reviewer for the positive comments and the suggestions, which helped to improve the manuscript. In the following we provide a point-to-point response to all reviewer comments. The reviewer's comments are printed in italic and our response in roman font type. We indicate the line numbers of the revised manuscript where larger revisions have been made. For the reviewer's convenience we also copied larger changes we made to the manuscript to this response and enclosed them with quotation marks.

*The manuscript presents improvements in a retrieval algorithm for ground-based thermodynamic profiles in the boundary layer. It is well structured and clear in the goals of the study as well as in the presentation of the results. I found the explanations and the illustration of the methodology well referenced and convincingly justified. I recommend the publication of the manuscript with just few minor/technical corrections.*

**Response:** Thank you for this positive evaluation.

**1.1 Minor comments**

1. *L61-63. I find this part a bit confusing as you just mentioned the need to inflate the noise and you apply a noise-reduction technique. Maybe to make the sentence clearer, I would reformulate it in this way: "The usage of the radiance uncertainty before noise filtering for the error covariance matrix together with the noise-filtered radiance in the measurement vector is intended to compensate for the missing forward model uncertainty".*

   **Response:** Thank you for this suggestion. We changed the text accordingly.

2. *L181-182. Can you clarify the usage of cloudy-contaminated data for the analysis? In particular, in the last paragraph of Sect. 2 it is not clear to me whether you use only cloud-free profiles for the IRS analysis or, as you said before in the manuscript, you keep cloud-free data only for the radiosonde comparison.*

   **Response:** For the statistical analysis in Figures 9, 11, 15, we completely excluded samples with cloud contamination for the IRS (all samples with $LWP > 8 \text{ g m}^{-2}$). When analyzing profiles, we allowed profiles with $LWP > 8 \text{ g m}^{-2}$, but did not use any data above cloud base height. This is

why the number of available data points decreases with height in Fig. 10. We rewrote this sentence to clarify (l. 191-193):

"This is why we excluded any profiles with LWP > 8 g m−2 in our statistical analysis for the IRS-based TROPoe experiments (Sects. 4.1 and 4.3). In our height-resolved analysis related to temporal consistency (Sect. 4.2.1), we excluded data above cloud base only instead of excluding the cloudy profiles completely."

3. *Two questions about the WVBAND experiment. Is the information about near-surface water vapor coming from Ymet? I understand that you use the information from the additional band according to the WV content in a linear fashion, but could this usage introduce an overall bias between dry air profiles and high-humidity profiles? Is this additional band used for all retrievals in Fig. 3 panel (c)? I notice that almost all values are changing in panel (c) with respect to panel (b), and I assume that the only change between the two is the additional band.*

**Response:** Yes, the information about near-surface water vapor is coming from Ymet (we added this information to the text). In the example in Fig. 3, the additional band in WVBAND is used in all samples, because the near-surface water vapor was above the threshold of 12 g kg$^{-1}$. This is the reason for the different values between NOISE (b) and WVBAND (c). The additional band is also used in TROPOEIN (see Table 2 for an overview of the configurations). We added this information to the text (l. 237-238):

"Since near-surface water vapor mixing ratio was above the threshold of 12 g kg$^{-1}$ throughout the day, the additional band is used in all profiles leading to slightly different values between NOISE and WVBAND."

To investigate if the additional band introduces a bias between dry and moist profiles, we searched for a period with a mix between dry (i.e., noise in the additional band largely inflated and thus not used) and moist conditions (i.e., noise not inflated and thus used). We were hoping to see if there is a jump between neighboring profiles that either used or did not use the additional band. However, the challenge is to find a day where moisture values spanned such a large range, i.e. below 7 and above 12 g kg$^{-1}$ in neighboring profiles. The best period we could find was at SGP with a rapid change in humidity related to a frontal passage (Fig. 1 in this response). Before the frontal passage on April 17, near-surface mixing ratio was close to 12 g kg$^{-1}$ (Fig. 1a), i.e. noise in the additional band was inflated only slightly (Fig. 1b). After the frontal passage shortly after midnight on April 18, humidity dropped and reached values of less than 7 g kg$^{-1}$, i.e. noise

in the additional band was strongly inflated. By comparing the time-height sections of NOISE (Fig. 1c) and WVBAND (1d), we were not able to identify more striping or biases between dry and moist profiles in WVBAND compared to NOISE, and therefore we do not believe that the inclusion of the additional WV band leads to a bias.

[Figure]

Fig. 1: (a) Near-surface mixing ratio used for inflating the noise in the band between 793 and 804 cm[-1], (b) noise at 800 cm[-1], (c) water vapor mixing ratio profiles in NOISE, and (d) water vapor mixing ratio profiles in WVBAND on April 17-18 2019 at SGP.

4. *Regarding the TROPOEIN experiment: is the usage of the additional information at a previous step in the measurement vector equivalent to using as a-priori information the retrieved profile at a previous time step? Or would this make the retrieval too tight to the previous state?*

**Response:** Using the retrieved profile as prior, could be an alternative way of including it in the retrieval. However, we prefer not to do this for two reasons. The first is as you suggest: it would be too restrictive to the previous state, especially in a covariance between levels perspective. The

second is philosophical: observations belong in the observation vector, and the prior should be only the climatology. We prefer this approach because then the denominator in the information content calculation stays the same, and thus we can more easily assess the improvement in the information content (i.e., the increase in the degrees of freedom for signal) when we use the TROPOEIN vs not.

**1.2 Technical comments**

*L7: is crucial → are crucial*

**Response:** Changed.

*L6-10: I suggest to move the sentence "The characterization of the uncertainty … for retrieval performance" right before "Since each profile…" and start here a new sentence "We present methods…"*

**Response:** Changed.

*L14: spectrometers, radiometers → spectrometer, radiometer*

**Response:** Changed.

*L16-17: I would reformulate as: "Observations of the continuous temporal evolution and the diurnal cycle of thermodynamic profiles are essential for the analysis of physical processes…."*

**Response:** Changed.

*L30: Shall you also spell AERIoe out?*

**Response:** AERIoe is essentially modeled after AERIprof – AERI being the instrument, and "oe" being the method used for the retrieval. We added the following information (l. 30-31):

"Based on the AERIoe optimal-estimation physical retrieval algorithm (Turner and Löhnert, 2014), which was developed for the Atmospheric Emitted Radiance Interferometers (AERI) instruments and only allowed infrared radiances as input, …"

*L46: I would replace "this process is iteratively repeated" with "the state vector is modified in an iterative process."*

**Response:** Changed.

*L76: spectral band from → spectral band at*

**Response:** Changed

*L95: are analyzed → is analyzed*

**Response:** Changed.

*L127: I would say "once daily only during intense observation periods"*

**Response:** Radiosondes were launched once daily every day of the campaign. During intensive observation periods, radiosondes were launched in up to 2 hour intervals. We changed the text to (l. 132-133):

'Radiosondes were launched twice per day at SMT and at least once daily, and more frequently during intensive observation periods, at SAV.'

*L127: numbers → number*

**Response:** Changed.

*L133: I think the detail about the usage of narrow and wide FOVs is possibly too technical, if you don't explain it further, I think it is better to just say that the detail of the usage of water vapor profiles is described in the papers you mention.*

**Response:** We removed this detail.

*L164. Isn't the 1-σ uncertainty the square root of the diagonal elements of the matrix Sop?*

**Response:** Yes. Changed.

*Fig.7 and 8: Since you first describe Fig.8 and then discuss Fig.7 (except for the reminder at L257) I would invert the order of the two figures.*

**Response:** We prefer to keep Fig. 7 before Fig. 8, because we think that the reference to the lines of the additive factor and multiplier in Fig. 7 in the paragraph (l. 254-262) is useful. Hence Fig. 7 is used before Fig. 8.

*End of Sect. 3, I would add a sentence informing that the results of the correlation analysis are presented in the next section.*

**Response:** Added.

L355: on the average → on average

**Response:** Changed.

*Caption of Fig.11: I find the term "probability" confusing; would it be appropriate to say "distribution of"?*

**Response:** Changed.

*L383: linesin → lines in*

**Response:** Changed.

*L384: lower → less relevant*

**Response:** Changed.

*L386: Like for → In the same way as for*

**Response:** Changed.

*L387: Please introduce again Fig.11 here, for example: "As reported in Fig. 11 bottom row, …"*

**Response:** Changed.

*L424: contain → have*

**Response:** Changed.

*L424: an additional spectral band → the additional spectral band*

**Response:** Changed.

*L425: add "ratio" to water vapor mixing*

**Response:** Changed.

*L427: I would delete the comma after "water vapor band"*

**Response:** Changed.

*L430: "10-min profile" → "10 minutes a profile" …*

**Response:** Changed.

*In the caption of Fig. A1, replace (Fig. A1)" with "(panel a)"*

**Response:** Changed.

---

## Author Comment (AC2)

**Response to comments of reviewer 2**

We thank the anonymous reviewer for the positive comments and the suggestions, which helped to improve the manuscript. In the following we provide a point-to-point response to all reviewer comments. The reviewer's comments are printed in italic and our response in roman font type. We indicate the line numbers of the revised manuscript where larger revisions have been made. For the reviewer's convenience we also copied larger changes we made to the manuscript to this response and enclosed them with quotation marks.

*This paper outlines three key improvements to the TROPoe retrieval algorithm, a software package that is seeing increasing use in both operational and research milieus. These three improvements, namely the addition of a water vapor band to the retrieval, the integration of the t–1 retrieval to help improve temporal consistency and reduce striping, and the impact of radiance noise inflation, are all discussed. Overall, this is a well-written and compelling manuscript that fits with the scope of AMT and is suitable for publication after a number of small issues are addressed. These are mostly issues associated with explanations and justifications.*

**Response:** Thank you for this positive evaluation.

*The most significant issue I see is in the conversation about inflating the radiance noise to account for the fact that the model error is not expressly addressed. This invokes a somewhat lengthly list of questions, but in its current form the manuscript could do more to justify why this approach is proper and valid. Are uncertainties really fungible like that? Can one inflate one set of uncertainties and assume that it encompasses a different set of uncertainties that exist for an entirely different set of reasons? What are the limitations on including model uncertainties in the retrieval (i.e. just how expensive is it to do it explicitly, and by what factor is the outlined approach better)? Has there ever been an attempt to treat the TROPoe model errors explicitly, and if so, how do those results compare to the noise inflation approach? What is the purpose of reducing the noise with the PCA filter if one is just going to inflate it right back up again? Why is the number of converged retrievals the appropriate measure to determine if the proper inflation factor has been reached?*

**Response:** The observational uncertainty (Se) needs to include contributions from the actual observations (Sy) and uncertainties in the forward model used to create the simulated observations (Sb'), using the

notations in Maahn et al. (2020). Specifying Sb' has been done for microwave profiling systems (i.e., Cimini et al. 2018); however, there are only a few dozen lines to consider in the microwave. There are approximately 1,000 water vapor absorption lines in the spectral region used for the TROPoe retrievals from infrared sounders. We would have to estimate the uncertainty in the strength, width, and temperature dependence of each (and how these uncertainties are correlated) to compute the uncertainty in the forward model properly. Thus, the dimension of the b parameter vector is approximately 3,000 elements, if we only consider water vapor. But there are nearly 20,000 $CO_2$ lines between 500 and 960 $cm^{-1}$ which would need to be considered also. Even if only the strongest absorption lines were considered, the total number of absorption lines (from $H_2O$ and $CO_2$) would still be more than 2,000 in total. We are working to do this in the infrared, but it is a big project and is work-in-progress.

And thus, we needed a way to account for the forward model uncertainty so that we don't overfit the data. Turner and Blumberg (2019) first used this approach of applying the noise filter to reduce the random error in the observations, but using the original observational uncertainty for the sum of the two components. We found that this approach is still insufficient for some instruments (the observational uncertainty is instrument dependent) and hence proposed a minimum noise level to be used. To determine the appropriate noise, we not only looked at the number of converged retrievals, but also considered cDFS and MAE compared to radiosondes. Our choice of the noise level was a compromise between solution availability, information content, and error.

We added a footnote to the introduction, describing the challenge with including the uncertainty of the forward model:

'There are approximately 1,000 water vapor lines and nearly 20,000 $CO_2$ lines in the spectral region used for TROPoe retrievals from IRS, and we would have to estimate the uncertainty in the strength, width, and temperature dependence of each (and how their uncertainties are correlated) to compute the uncertainty in the forward model. Even if only the strongest lines were included, the number of lines would still exceed 2,000 in total. For MWR, the uncertainty of the forward model has been specified by Cimini et al. (2018); however, there are only a few dozen lines to consider in the microwave.'

*1    Some other smaller issues:*

*In many cases, the IRSes and MWRs used in this study are not at the same location, but instead are located within the same general climate regime. Does that have any impact, i.e.can we compare the*

*moisture variability for the tropical IRS to that of the MWR? My guess is that it's fine, but it probably should be discussed.*

**Response:** In Fig. 2, we show near-surface temperature over mixing ratio for all sites to illustrate the differences in climatological conditions. Mixing ratio values at the two tropical sites (MAO and SAV) and at the two mid-latitude sites (SGP in April and LIN) are in a similar range. However, the variability is different. To better illustrate this, we added mean and standard deviation for each site to Fig. 2. For example, moisture variability was larger at SGP in April than LIN and larger at MAO than SAV, indicated by the larger errorbars. Because TROPOEIN aims to improve the temporal consistency in the high-pass filtered TROPoe retrievals (using a cut-off time of 3 h), we compared standard deviation for water vapor and temperature for high-pass and low-pass filtered data separately (Figs. 1,2 in this response). Most of the difference in variability between the sites in the same climatological regime are found in the low-pass filtered data (blue bars). The variability in the high-pass filtered data is much more similar, which means that the improvements of temporal consistency in the TROPOEIN experiments are done for similar conditions and that the findings for IRS- and MWR-based retrievals are comparable.

[Figure]

Figure 1: Standard deviation of near-surface measured water vapor mixing ratio for high and low pass-filtered data.

[Figure]

Figure 2: Standard deviation of near-surface measured temperature for high and low pass-filtered data.

We added this sentence to the description of thermodynamic conditions at the sites (l. 114-119):

"While the mean values are very similar for sites in the same climatological regime, i.e. MAO and SAV and SGP in April and LIN, the standard deviations vary, which may have implications for our experiment to improve the temporal consistency. However, the differences in standard deviation are mostly related to variations on time scales of several hours and more. Since we evaluate the improvements to temporal consistency on a shorter time scale, we are confident that the results for IRS- and MWR-based retrievals in the same climatological regime are comparable."

*Line 119: is it "fore optics," "fore-optics," or "foreoptics?" I've seen all three, but I think I've seen the last one the most.*

**Response:** Changed to foreoptics.

*Line 129, Table 1: The geographic column would benefit by also adding some place names ("Oklahoma USA," "Brazil," "Greenland," etc.) Also, the parentheses in the Number of Radiosondes column are mismatched.*

**Response:** Added and changed.

*Line 158: Specify that the 55 levels in thermodynamic profile retrievals are for TROPoe; as it is written, it sounds like it's the case for all thermodynamic retrievals regardless of instrument.*

**Response:** Changed.

*Lines 216-226: Adding the WV band may help the number of converged retrievals, but is there an impact on their accuracy? Moreover, is there a discernable impact on the performance of the T retrievals in addition to the WV retrievals?*

**Response:** The accuracy of the retrievals is investigated by comparing the retrieved profiles to radiosonde profiles in Sect. 4.3. Mean absolute and relative errors are shown in Fig. 15. Adding the additional water vapor band improved the retrieval errors for both water vapor and temperature. The improvements in temperature retrievals were smaller than for water vapor retrievals. This is described in Sect. 4.3

*Line 251: The temporal consistency between the atmosphere is going to vary based on the diurnal cycle. Are there plans to vary the noise inflation uncertainty of the previous retrieval based on time of day?*

**Response:** This is a great thought. We currently have no plans to implement a time dependent noise inflation uncertainty, but may consider it in the future.

*Lines 255-259: how were the specific values for N decided?*

**Response:** We chose the values for N empirically. These values need to be large enough to not suppress any real variability in the boundary layer between the 10-min consecutive profiles. The chosen values for N increase the uncertainty of water vapor in the boundary layer by a factor of up to 5 and increase the uncertainty of temperature by up 3 deg. The typical uncertainty in the boundary layer for water vapor is between 0.5 and 1 g/kg (except for the very dry environment at SMT) which means that water vapor mixing ratio in the boundary layer is allowed to change by more than 2.5 g/kg in a 10 min period without constraining the solution by the previous profile. We decided to go with these rather high values for N to be on the conservative side.

We added an explanation to the text (l.273-275):

"The values of N were determined empirically and the rather high values in the boundary layer allow water vapor mixing ratio to change by more than 2.5 g kg$^{-1}$ (with $\sigma_{WVMR}$ typically larger than 0.5 g kg$^{-1}$) and temperature by more than 3 °C close to the surface within a 10 min period, without suppressing the change by the previous profile. "

*Line 276: If the processing is typically executed independently for each day, then when looking at continuous time series that span the 0000 UTC hour, there will be an artifact of increased variability for some discernible time period every day. Can the algorithm be modified to take into account retrievals from the previous day?*

**Response:** The reviewer is right that the profiles shortly after 0 UTC are impacted by running the retrieval for individual days and will always have a lower information content. Because TROPoe is computationally expensive (especially for the IRS-based retrievals), several days are usually processed in parallel when historical data are being processed. This means that conditions of the previous day before midnight are not necessarily available when the retrieval is run for a specific day. But in a real-time processing mode, the code could be modified to read in the output from the previous day so that this artifact won't exist. We added this information to the text (l. 292-296):

'Note that this independent processing of individual days may lead to an artifact of increased variability shortly after 00:00 UTC. The independent processing is done because TROPoe is computationally expensive (especially for the IRS-based retrievals) and the retrieval is usually run for several days in parallel when historical data are being processed. In real-time processing this artifact could be avoided by reading in the output from the previous day. '

*Line 368: fewer, not less*

**Response:** Changed.

*Line 470: Remove the comma between noise and decreases.*

**Response:** Changed.

**References**

Cimini, D., Rosenkranz, P.W., Tretyakov, M.Y., Koshelev, M.A. and Romano, F., 2018. Uncertainty of atmospheric microwave absorption model: impact on ground-based radiometer simulations and retrievals. *Atmospheric Chemistry and Physics*, *18*(20), pp.15231-15259.

Maahn, M., Turner, D.D., Löhnert, U., Posselt, D.J., Ebell, K., Mace, G.G. and Comstock, J.M., 2020. Optimal estimation retrievals and their uncertainties: What every atmospheric scientist should know. *Bulletin of the American Meteorological Society*, *101*(9), pp.E1512-E1523.

---

## Author Response (AR2)

**Second review round for manuscript 'Improving solution availability and temporal consistency of an optimal estimation physical retrieval for ground-based thermodynamic boundary layer profiling'**

**Response to Reviewer 2**

We appreciate the reviewer's comment which made us think again on how we can better explain the noise inflation approach and hope that our response is satisfying for the reviewer. The reviewer's comment is printed in italic and our response in roman font type.

*I am largely satisfied with the changes that have been made to the manuscript. I do feel that a little more discussion on the noise is appropriate: specifically, why are you filtering noise out with the PCA noise filter only to increase it back again? Perhaps I am not understanding it the way it is written. This is a one-to-two sentence change, and so I am satisfied with a minor revision of this point.*

We modified the text in the introduction trying to better explain how and why we need to inflate the uncertainty:

"Ideally, uncertainties in the observations, prior, and forward model are propagated and characterized by the posterior covariance matrix which is part of the TROPoe output. Because including the uncertainty of the forward model would increase the computational costs of the retrieval substantially, the uncertainty of the forward model is assumed to be zero in the current framework of TROPoe. Instead, the missing uncertainty of the forward model is assumed to be included in the uncertainty of the infrared radiances in the error covariance matrix of the observations. The uncertainty of the infrared radiances is instrument specific and is determined during the IRS calibration process (see Revercomb et al. (1988) and Knuteson et al. (2004b) for details). A common approach is to greatly reduce the random noise of the infrared radiances using a principal component-based noise filter before the radiances are used within TROPoe (Turner et al., 2006). At the same time, the original radiance uncertainty is included in the error covariance matrix of the observations of the retrieval. The intention is that the larger original radiance uncertainty captures the sum of the lower uncertainty of the noise-filtered radiances and the forward model uncertainty. For details on this approach see Turner and Blumberg (2019). However, depending on the radiance noise level of a specific IRS, the original radiance uncertainty might not be sufficient to compensate for the missing uncertainty of the forward model for some instruments, which may still lead to overfitting of the data and unrealistic profiles (Adler et al., 2023). We propose a minimum noise level for infrared radiances which should be used for the IRS radiance uncertainty in TROPoe as an intermediate solution before a computationally efficient implementation of the IRS forward model error can be included in the TROPoe framework. Because the signal to noise ratio in the MWR brightness temperature observations is lower than for the IRS radiances, overfitting is less of an issue for MWR-based retrievals."